# Modulation of insulin secretion by RBFOX2-mediated alternative splicing

Nicole D. Moss[1], Kristen L. Wells [1], Alexandra Theis[1], Yong-Kyung Kim [1], Aliya F. Spigelman [2], Xiong Liu [2], Patrick E. MacDonald[2] & Lori Sussel [1] ✉

Insulin secretion is a tightly regulated process that is vital for maintaining blood glucose homeostasis. Although the molecular components of insulin granule trafficking and secretion are well established, how they are regulated to rapidly fine-tune secretion in response to changing environmental conditions is not well characterized. Recent studies have determined that dysregulation of RNA-binding proteins (RBPs) and aberrant mRNA splicing occurs at the onset of diabetes. We demonstrate that the RBP, RBFOX2, is a critical regulator of insulin secretion through the alternative splicing of genes required for insulin granule docking and exocytosis. Conditional mutation of *Rbfox2* in the mouse pancreas results in decreased insulin secretion and impaired blood glucose homeostasis. Consistent with defects in secretion, we observe reduced insulin granule docking and corresponding splicing defects in the SNARE complex components. These findings identify an additional mechanism for modulating insulin secretion in both healthy and dysfunctional pancreatic β cells.

Diabetes is one of the fastest growing diseases and a leading cause of death globally[1]. Fundamentally, diabetes is characterized by the inability to maintain blood glucose homeostasis due to loss and/or dysfunction of insulin-producing β cells. A critical aspect of β cell function that is affected in diabetes is the ability to secrete insulin in response to elevated blood glucose levels. In healthy β cells, insulin is packaged into specialized vesicles known as granules. Insulin granules can be broadly categorized into two subpopulations, docked and undocked, in part based on their subcellular localization. Approximately 10% of insulin granules are docked and poised on the plasma membrane, including ~1% that are referred to as ready-releasable[2-4]. These granules are released as an initial and rapid response to changes in blood glucose levels in the first phase of biphasic insulin secretion. The availability of poised and docked granules is a limiting factor in maintaining blood glucose homeostasis through sustained insulin secretion[5].

In humans, the dramatic decrease in first-phase insulin secretion is early evidence for pre-diabetes[6,7] and is later coupled with a decrease in second-phase insulin secretion in type 2 diabetes (T2D)[8]. The decreased insulin secretion can be attributed to disruption in insulin granule distribution, docking, and exocytosis[5,9,10]. Insulin granule biogenesis, maturation, and sorting occurs in the cytoplasm. Mature granules destined for the plasma membrane are transported along microtubules[11]. Once at the membrane, granules are docked through interactions with the SNARE complex and other factors that help regulate exocytosis[12]. Evidence from studies in humans, and rat and mouse models of diabetes indicate that β cells have disrupted exocytosis and decreased expression of several exocytosis-related proteins, including components of the SNARE complex[6,8,13–16].

In addition to disrupted gene expression, there is emerging evidence implicating RNA-binding protein (RBPs) and aberrant splicing activity in the pathogenesis of diabetes and maintenance of β cell function[17–22]. RBPs bind to target RNAs to direct a host of functions, including the regulation of alternative splicing. Alternative splicing is a tightly regulated molecular process by which exons of a specific transcript are linked in different combinations to generate distinct isoforms of a single gene. This process often occurs in a cell-specific or context-dependent manner. RBPs can bind near intron-exon boundaries to

[1]Barbara Davis Center for Diabetes, University of Colorado Anschutz Medical Campus, Aurora, CO, USA. [2]Department of Pharmacology and Alberta Diabetes Institute, University of Alberta, Edmonton, AB, Canada. ✉e-mail: lori.sussel@cuanschutz.edu

block or recruit splicing factors like the spliceosome or other RBPs. Each RBP can have hundreds of targets within a cell[23–25], making their potential impact on cellular identity and function extensive. Here we identify RBFOX2 as a critical regulator of alternative splicing in the pancreatic β cell. RBFOX2 is a member of the RBFOX family of RBPs which predominantly regulate alternative splicing events by binding to a (U)GCAUG consensus sequence near alternative exons[26,27]. In other cell types and systems, RBFOX2 has been shown to coordinate interactions between pre-mRNA and processing machinery by recruiting or blocking the spliceosome at alternative splice sites among other things[28]. RBFOX2 is also implicated in a wide variety of biological processes including stem cell differentiation[29], development[29–31], mature cell function[28], stress response[32], and disease[21,33–35].

In this study, we define the role of RBFOX2 as a key modulator of insulin secretion through the regulation of alternative splicing of insulin exocytosis-related factors. Our analysis confirms *Rbfox2* expression is decreased in mouse models and humans with diabetes and shows that many putative RBFOX2 targets are mis-spliced in disease. We present evidence that independent of disease state, RBFOX2 is required to regulate insulin secretion through the subcellular organization and docking of insulin granules. In the absence of *Rbfox2*, insulin granules are not properly docked on the plasma membrane and insulin secretion is decreased. Consistent with its role as a splicing regulator, we show that RBFOX2 binds directly downstream of its target exons to promote exon inclusion. Together these data support an additional mechanism for fine-tuning insulin granule exocytosis through the regulation of alternative splicing by RBFOX2.

## Results

### Diabetes onset results in significant changes in alternative splicing and *Rbfox2* expression

RNA-binding proteins (RBPs) and RNA splicing events have been shown to be dysregulated in the islets of mouse models and humans with diabetes[17–21]. To begin to understand the roles of these RBPs in β cell function and diabetes, we sought to identify candidate RBPs in models of type 2 diabetes (T2D). Reanalysis of a comprehensive dataset published by Wihelmi et al.[21] that compared gene expression in islets from adult mice with obesity but not diabetic carrying a homozygous mutation in the leptin gene (*Lep^ob^, ob/ob*, ND) to adult mice with obesity, hyperglycemia, and insulin resistance (New Zealand Obese, *NZO*, T2D) confirmed 1973 significant differential splicing events across 1,339 genes in diabetic conditions (Fig. S1A). The majority of these differential splicing events are classified as cassette or skipped exon (SE, 65.69%) (Fig. S1B, Supplementary Data 1–6).

Corresponding to the changes in splicing events, there were many differentially expressed candidate RBPs that potentially regulate diabetes-sensitive splicing events. Of the most highly and differentially expressed RBPs, 17 RBPs had significantly decreased expression and 33 RBPs had increased expression in the T2D (*NZO*) compared to the ND (*ob/ob*) controls (Fig. 1A and Supplementary Data 7). As previously reported, one of the top downregulated genes was *Rbfox2* (Fig. 1B), a member of the RBFOX family of RBPs that have been implicated in the regulation of alternative splicing in a variety of cell types[29,31–33,36] and several disease models[19–21,24,34–37]. In both mice and humans, *Rbfox2* is highly expressed in pancreatic β

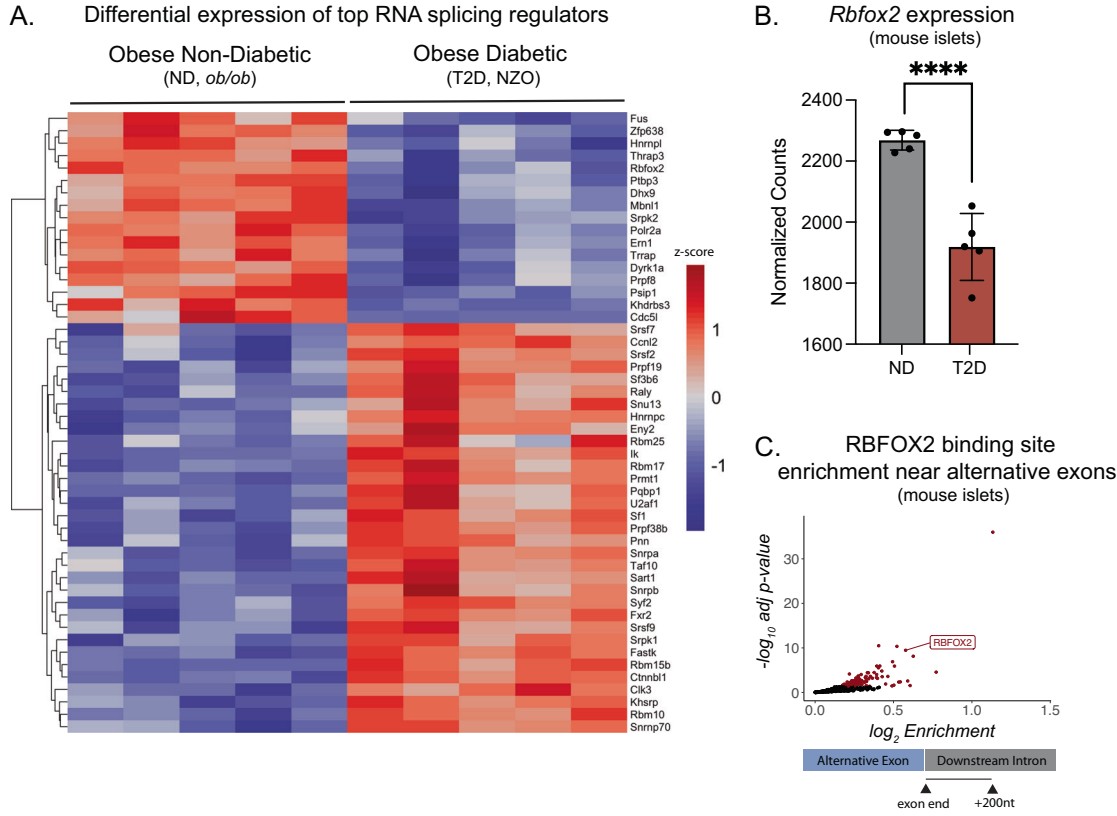

**Fig. 1 | *Rbfox2* expression is dysregulated in diabetes and the RBFOX2 consensus sequence is enriched near alternative exons. A** Mean-centered, normalized expression of most highly expressed splicing regulators in islets from mice with obesity but without diabetes (ND, *ob/ob*, n = 5) and islets from mice with both obesity and diabetes (T2D, NZO, n = 5) mouse islets from RNA-seq (GSE183247). **B** Normalized read counts of *Rbfox2* in ND (*ob/ob*) and T2D (NZO) mouse islets, statistical analysis by DESeq2 p-adj = 0.00914. **C** 5-mer enrichment 200nt

downstream of alternative cassette exons identified from rMATS (FDR < 0.05, ΔPSI > 0.01) in ND (*ob/ob*, n = 5) and T2D (NZO, n = 5) mouse islets, enrichment relative to constitutive exons plotted adj *p*-value from binomial test with Bonferroni correction, each dot represents a unique 5-mer, RBFOX2 5-mer GCAUG is labeled (GSE183247). (Data are represented as mean values with SD, ns *p* > 0.05, **p* ≤ 0.05, ***p* ≤ 0.01, ****p* ≤ 0.001, *****p* ≤ 0.0001).

cells and other endocrine cell types of the islet (Figs. S1D, S2A) but is not expressed in adult exocrine tissue[38,39]. Analysis of several published mouse and human islet single-cell data sets demonstrated that *Rbfox2* is the only member of the RBFOX family member expressed in β cells and whose expression changes under diabetic stress (Fig. S1C). Notably, although there are few available complementary datasets, *RBFOX2* mean expression was also decreased in human islets from donors with T2D islets (Fig. S2B)[40] and sorted human β cells from donors with T2D (Fig. S2C)[41].

To identify whether the presence of an RBFOX2 consensus sequence, GCAUG[27], correlated with alternative splicing events, we used a sliding window analysis to identify all combinations of 5 nucleotides (5-mers) near the alternatively spliced cassette exons (SE) present in the T2D samples compared to the ND controls. We determined that the RBFOX2 consensus binding sequence is significantly enriched downstream of the differentially expressed exons (Fig. 1C, Supplementary Data 8), suggesting that differential expression of RBFOX2 could directly contribute to alternative splicing events in this mouse model of T2D. Furthermore, alternative splicing analysis of datasets collected from human islet samples (GSE164416)[40], demonstrated that the RBFOX2 consensus sequence was enriched downstream of cassette exons that were differentially included in islets from individuals with T2D (Fig. S2D, Supplementary Data 8). Together these data support RBFOX2 as a strong candidate for regulating alternative splicing in diabetes and maintaining endocrine cell health and function.

## Conditional loss of RBFOX2 from mouse pancreas results in dysregulation of glucose homeostasis

To determine whether RBFOX2 is required to maintain the health and function of pancreatic endocrine cells, we generated a pancreas-specific *Rbfox2*-mutant mouse using the *Pdx1*:CRE allele[42] and two copies of a floxed *Rbfox2* allele[30], hereafter referred to as *Rbfox2*-mut mice (Fig. 2A). The pancreas-specific *Rbfox2*-mut mice were born with normal body weight (Fig. S3A) and *ad libitum* blood glucose levels (Fig. S3B). The mice developed normally and there were minimal differences in weight or fasting blood glucose levels at 4 and 8 weeks of age (Fig. S3C–F). Histological analysis also demonstrated that *Rbfox2*-mut mice had appropriately formed islets with β cells clustered in the interior and the other endocrine cells around the perimeter (Fig. 2B). Additionally, there were no significant quantitative differences in insulin area or C-peptide area (Fig. 2C, D). Expression of islet endocrine hormones, including insulin (*Ins1, Ins2*) and glucagon (*Gcg*) were unchanged although there was a modest decrease in somatostatin (*Sst*) and pancreatic polypeptide (*Ppy*) expression, although this did not correlate with a change in the pan endocrine marker Chromogranin A (*Chga*) (Fig. 2E).

Despite the apparent normal formation of the pancreatic islets, intraperitoneal glucose tolerance tests (IP-GTT) showed that *Rbfox2*-mut mice were glucose intolerant as early as 4 weeks of age (Fig. S3G–J), although, similar to many mouse models of diabetes, the phenotype was more severe in male mice. Glucose homeostasis remained impaired throughout life with both sexes becoming significantly glucose intolerant by 8 weeks of age (Figs. 2F, G, S3K–N). Heterozygous *Rbfox2*-mut mice were euglycemic at all ages tested (Figs. 2F, G, S3G, H). Additionally, oral glucose tolerance tests (O-GTT) demonstrated that there were no significant differences between the *Rbfox2*-mut mice and controls (Fig. 2H, I). The IP-GTT measures the β cell response to direct glucose stimulation and can more readily identify defects in first phase insulin secretion, whereas the O-GTT assay assesses contribution of GLP-1 signaling from the intestine and can reveal second phase secretion defects. Together these data suggest that *Rbfox2*-mut mice have defects that were intrinsic to β cell function, predominantly in the first phase of insulin secretion response.

## Pancreatic deletion of *Rbfox2* disrupts splicing of insulin secretion pathway genes

To understand how changes in alternative splicing in *Rbfox2*-mut islets could disrupt β cell function, we performed transcriptome and Multivariate Analysis of Transcript Splicing (rMATs) analysis on islets from *Rbfox2*-mut mice and their littermate non-CRE controls to identify 878 splicing events in 610 differentially spliced genes (Fig. S4A and Supplementary Data 9–13). To distinguish β cell-specific altered splicing events and to establish a cell system that would allow us to probe the molecular activity of RBFOX2, we also took a siRNA approach to knockdown *Rbfox2* in the mouse-insulinoma cell line (MIN6) (Fig. 3A). We were able to achieve ~70% knockdown of the *Rbfox2* mRNA (Fig. 3B) and a corresponding reduction in RBFOX2 protein (Figs. 3C, S4B). We performed a similar analysis using rMATs to identify alternative splicing in β cells and identified 872 alternative splicing events across 606 genes (Fig. S4A). Notably, there was no increase in expression of the other RBFOX family members, *Rbfox1 and Rbfox3*, to potentially compensate for the loss of *Rbfox2* in the *Rbfox2*-KD cells or in the *Rbfox2*-mut islets (Figs. 3D, S4C).

Comparison of the differential RNA splicing events in *Rbfox2*-mut islets and *Rbfox2*-KD MIN6 cells revealed a significant overlap of alternatively spliced genes between the two datasets across a range of ΔPSI (change in percent spliced in) thresholds (Fig. S5A), suggesting that many of the splicing changes in the mouse islet can be attributed to splicing changes in the β cell. Additionally, *Rbfox2*-KD cells exhibited similar changes in overall splicing patterns with cassette/skipped (SE) and mutually exclusive exons (MXE) being the most abundant (Fig. S5B). This analysis also allowed us to determine the proportion of alternative splicing events observed in obese diabetic mice (T2D, *NZO*) that were potentially mediated by RBFOX2 (Supplementary Data 14). Our data indicates a high level of overlap at the gene level. We further investigated the correlation of alternative splicing events at the exon level and observe that there are a subset of alternative splicing events, shared between the *Rbfox2*-KD β cells and *Rbfox2*-mut islets (Fig. S5C), as well as the *Rbfox2*-mut islets and T2D islets (Fig. S5D). Notably, there were 196 genes significantly alternatively spliced across all three datasets at a threshold of 1% ΔPSI and 112 at 5% ΔPSI (Figs. 3E and S5E), and gene ontology (GO) analysis suggested that there was an enrichment of these genes in pathways relating to mediating exocytosis and microtubule-based sub-cellular organization (Figs. 3F, G, S5F). This includes components of the SNARE complex – *Snap25, Stxbp1, Syt7*, and microtubule associated proteins – *Dcnt1, Clip1, Mapt*, among others (Fig. 3G, red text).

## Loss of RBFOX2 disrupts subcellular insulin granule organization in pancreatic β cells

Many of the alternatively spliced genes identified as sensitive to RBFOX2 dependent alternative splicing were involved in vesicle trafficking and insulin granule docking. To determine whether the changes in the alternative splicing of these genes in *Rbfox2*-mut islets result in phenotypic changes in subcellular granule organization, we imaged islets using electron microscopy (EM). To examine the granule organization, we collected EM images at 30,000x magnification and quantified equivalent cytoplasmic area and depth from the membrane across both the *Rbfox2*-mut and control β cells (Figs. 4A, S6A, B). We observed an increase in insulin granule diameter (Fig. S6C). Additionally, we observed a change in granule distribution with fewer granules within 50 nm of the plasma membrane in the *Rbfox2*-mut β cells despite no differences in insulin granule density (Figs. 4B, S4D). Quantification of insulin granule numbers above and below the 50 nm threshold, which approximated docked vs. undocked granules, revealed there was a significant decrease in the number of docked granules in the *Rbfox2*-mut compared to the control (Fig. 4C). These data suggest that RBFOX2 is critically important for the splicing of genes involved in insulin granule organization near the plasma

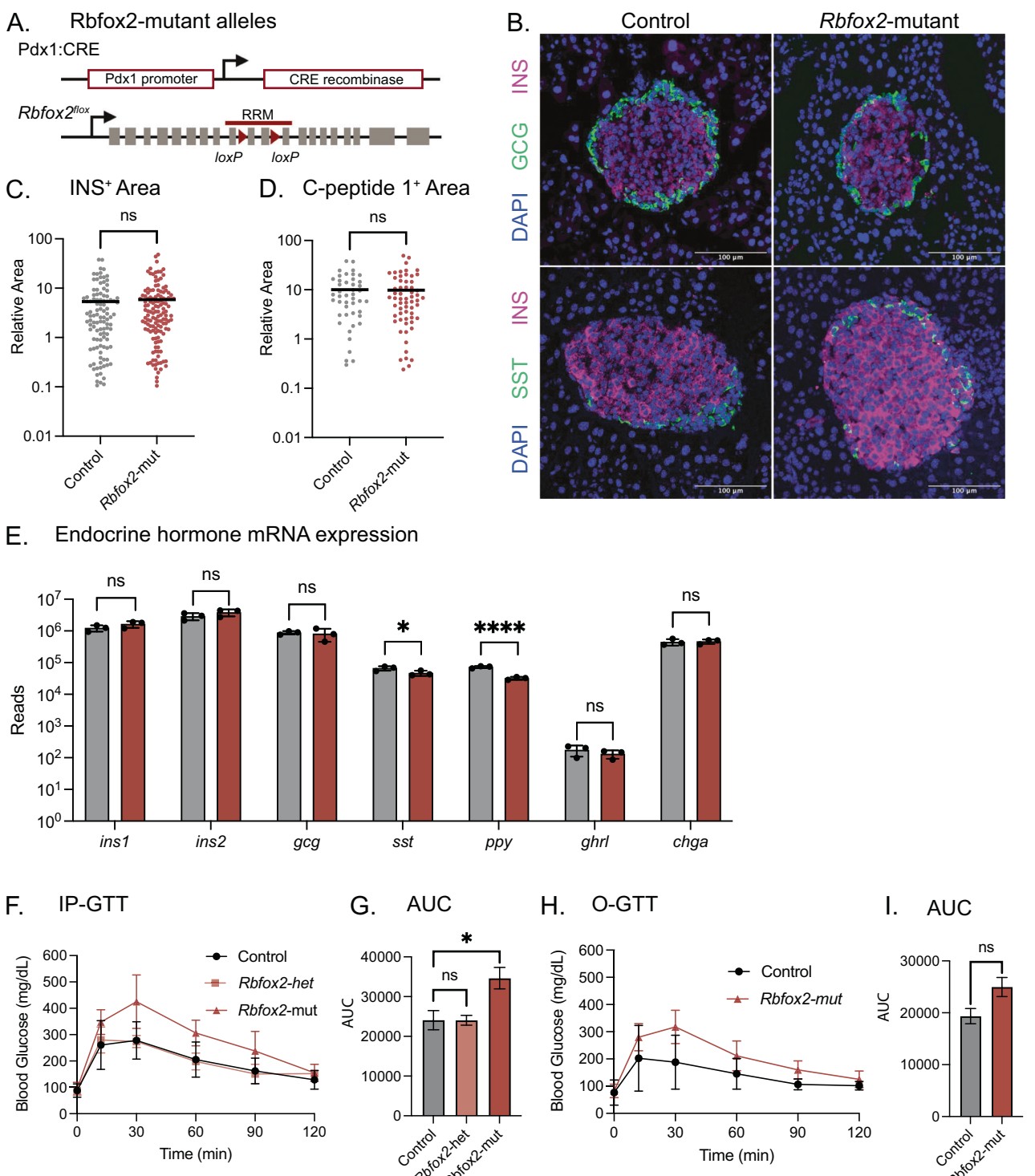

**Fig. 2 | Conditional *Rbfox2* mutant mice are glucose intolerant. A** Pancreas specific *Rbfox2*-mutant mouse alleles, including *loxP* sites flanking the RNA recognition motif (RRM) in the *Rbfox2* allele. **B** Immunofluorescence showing staining for DAPI (blue) and labeled with anti-insulin (INS, magenta) and anti-glucagon (GCG, green) or anti-somatostatin (SST, green) in control (*Rbfox2*fl/fl) and *Rbfox2-mut* (*Pdx1*:CRE; *Rbfox2*fl/fl) islets. **C** Quantification of anti-insulin⁺ area normalized by islet area, *n* = 3 male mice per group, *t*-test. **D** Quantification of anti-C-peptide 1⁺ area normalized by islet area, *n* = 3 male mice per group, *t*-test. **E** RNA expression of endocrine hormones from isolated islets of 8-week-old mice, control (*Rbfox2*fl/fl, *n* = 2 male, 1 female) and *Rbfox2-mut* (*Pdx1*:CRE; *Rbfox2*fl/fl, *n* = 2 male, 1 female),

significance determined by DESeq2 (*ins1* padj = 0.99, *ins2* padj = 0.99, *gcg* padj = 0.99, *sst* padj = 0.049, *ppy* padj = 8.8 × 10⁻¹³, *ghrl* padj = 0.99, *chga* padj = 0.99) (**F**) Intraperitoneal glucose tolerance test (IP-GTT) in 8-week male mice, control (*Rbfox2*fl/fl or *Rbfox2*fl/+, *n* = 7), *Rbfox2-het* (*Pdx1*:CRE; *Rbfox2*fl/+, *n* = 3) and *Rbfox2-mut* (*Pdx1*:CRE; *Rbfox2*fl/fl, *n* = 6). (**G**) Area under the curve (AUC) and SD for data presented in (**F**), one-way ANOVA with multiple comparisons *F* = 6.479. **H** Oral glucose tolerance test (O-GTT) in 8-week male mice, control (*Rbfox2*fl/fl, *n* = 5) and *Rbfox2-mut* (*Pdx1*:CRE; *Rbfox2*fl/fl, *n* = 7). **I** Area under the curve (AUC) and SD for data presented in (**H**), *t*-test two-tailed *t* = 2.066, df = 9. (Data are represented as mean values with SD, ns *p* > 0.05, **p* ≤ 0.05, ***p* ≤ 0.01, ****p* ≤ 0.001, *****p* < 0.0001).

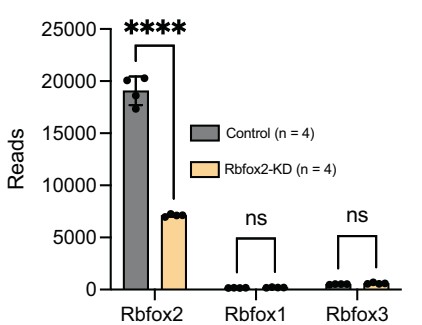

**A.** Rbfox2 knockdown strategy

**D.** *Rbfox expression*

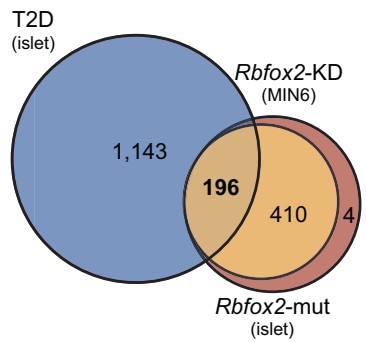

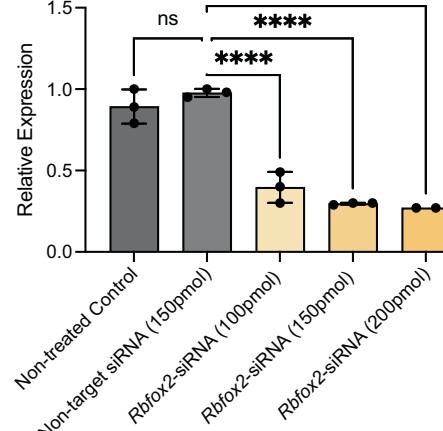

**B.** Rbfox2 mRNA

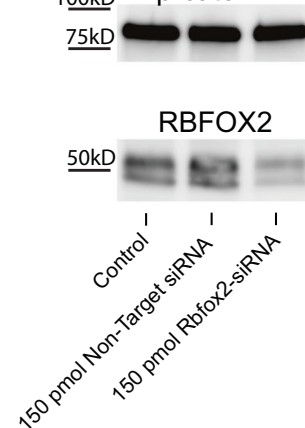

**C.** RBFOX2 protein

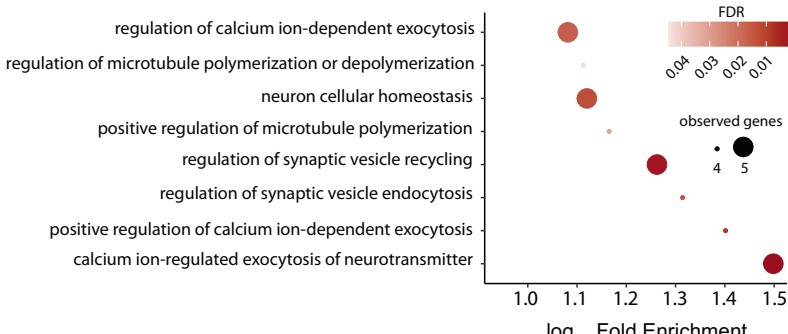

**E.** Overlapping alternatively spliced genes

**F.** Top GO Terms for overlapping genes

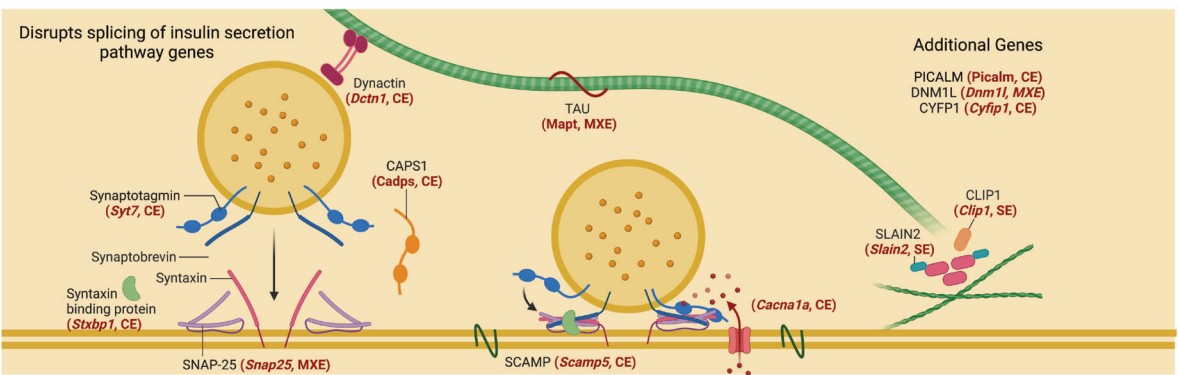

**G.** RBFOX2 sensitive genes involved in insulin secretion

**Fig. 3 | Alternative splicing of insulin secretion machinery in β cell specific knockdown of *Rbfox2*. A** *Rbfox2* siRNA targeting strategy in MIN6 cells. **B** qRT-PCR measuring *Rbfox2* mRNA following siRNA titration normalized to actin, $n = 3$ for each condition, one-way ANOVA with multiple comparisons $F = 70.52$. **C** RBFOX2 western blot following 48-h 150 pmol siRNA treatment ($n = 1$). **D** *Rbfox2* expression by RNA-Seq, $n = 4$ for each group, statistical significance determined by DESeq2 (*Rbfox2* padj $= 4.12 \times 10^{-203}$, *Rbfox1* padj $= 0.28$, *Rbfox3* padj $= 0.27$). **E** Intersection of significantly alternatively spliced genes (FDR $< 0.05$, |ΔPSI| $> 0.01$) in *Rbfox2*-KD in

MIN6 cells (yellow) with *Rbfox2*-mut mouse islet (red) and T2D mouse islet (*ob/ob* v NZO, blue). **F** Top terms from GO Term analysis of 196 overlapping genes. **G** Diagram of function for selected genes in top GO Terms, red gene names are significantly alternatively spliced in either cassette/skipped exon (SE) or mutually exclusive exon (MXE) across datasets (created with BioRender.com). (Data are represented as mean values with SD, ns $p > 0.05$, $^*p \leq 0.05$, $^{**}p \leq 0.01$, $^{***}p \leq 0.001$, $^{****}p < 0.0001$).

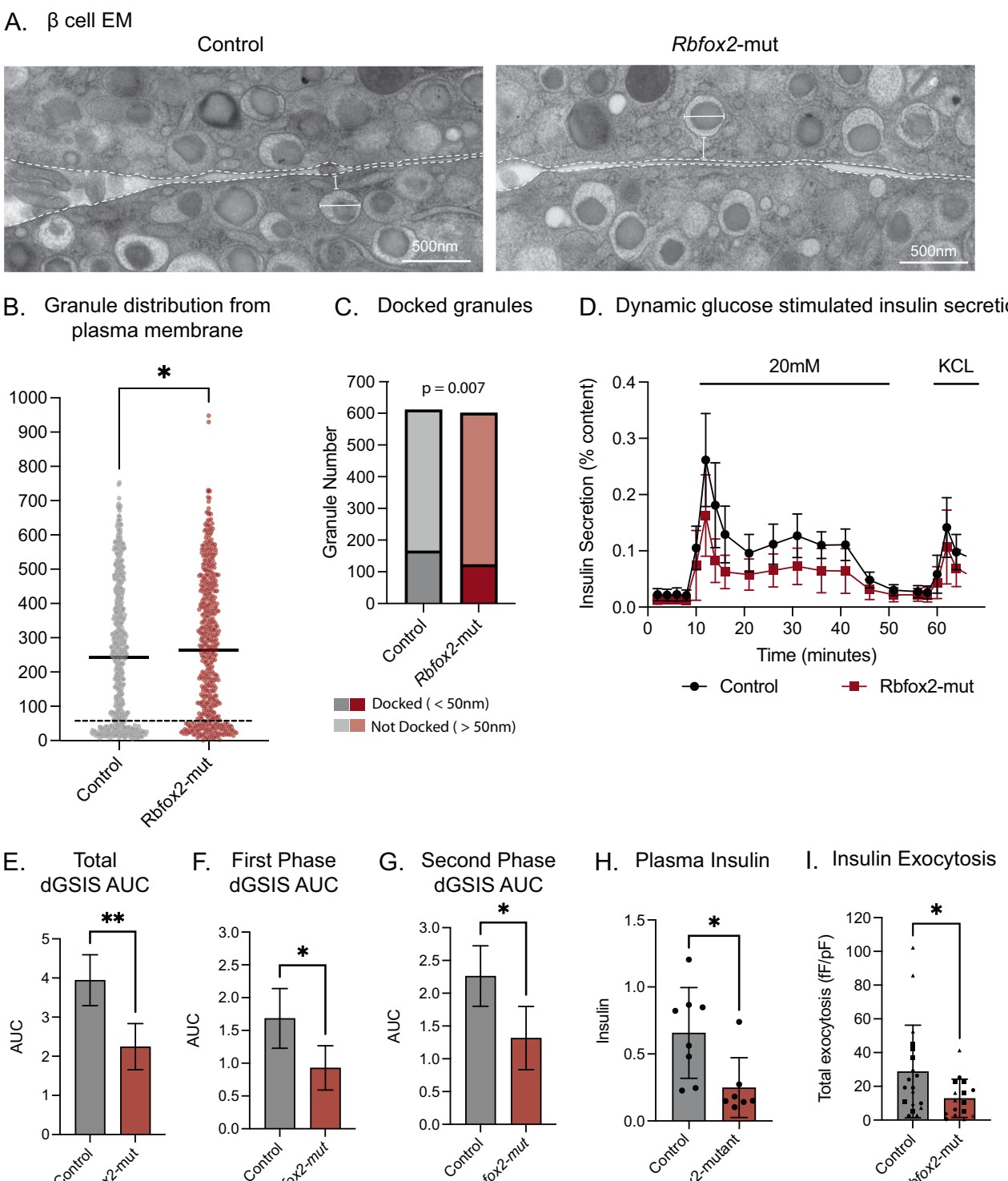

**Fig. 4 | Conditional *Rbfox2* mutation results in decreased insulin secretion and reduction in docked insulin granules. A** Representative EM image of adjacent β cells. **B** Quantification of distance from granule edge to plasma membrane, $n = 3$ mice per group and each point is a granule, $p = 0.02$ t-test two-tailed $t = 2.313$, df = 1210. **C** Quantification of docked and undocked granules, $n = 3$ mice per group, statistical comparison by Fisher's Exact Test. **D** Dynamic glucose stimulated insulin secretion (dGSIS) assay measuring insulin secretion as a percent of total insulin content in $n = 4$ control and $n = 4$ *Rbfox2*-mut mice. **E** Quantification of dGSIS assay comparing area under the curve, $p = 0.0082$, mean AUC and SD from data presented in (**D**) t-test two-tailed $t = 3.88$, df = 6. **F** Quantification of dGSIS assay comparing area under the curve during first phase insulin secretion from $t = 10$ to

$t = 21$, $p = 0.0376$, mean AUC and SD from data presented in (**D**) t-test two-tailed $t = 2.659$, df = 6. **G** Quantification of dGSIS assay comparing area under the curve during second phase insulin secretion from $t = 21$ to $t = 41$, $p = 0.0297$, mean AUC and SD from data presented in (**D**) t-test two-tailed $t = 2.2836$, df = 6. **H** Plasma insulin measured by ELISA 15 min post glucose injection, $n = 8$ control and $n = 7$ mutant mice, $p = 0.0178$ t-test two-tailed $t = 2.712$, df = 13. **I** Insulin exocytosis following membrane depolarization, $n = 19$ cells across 4 control biological replicates and $n = 16$ cells across 4 *Rbfox2*-mut biological replicates, $p = 0.0376$, statistical significance determined by t-test two-tailed $t = 2.166$, df = 33. (Data are represented as mean values with SD, ns $p > 0.05$, *$p \leq 0.05$, **$p \leq 0.01$, ***$p \leq 0.001$, ****$p < 0.00001$).

membrane and that deletion of *Rbfox2* contributes to defects in rapid response to changes in blood glucose by disrupting granule docking and ultimately insulin secretion.

The disrupted glucose homeostasis phenotype of the *Rbfox2*-mut mice combined with the overwhelming number of alternatively spliced genes involved in insulin secretion and the observed defects in granule docking prompted us to assess whether the *Rbfox2*-mut islets were defective for insulin secretion. Dynamic glucose stimulated insulin secretion assays (dGSIS) on isolated islets demonstrated a significant defect in insulin secretion at high glucose concentrations (Fig. 4D, E). The *Rbfox2*-mut islets had decreased insulin secretion during both the first and second phase of dGSIS as measured by the area under the curve (AUC) (Fig. 4F, G). We further confirmed decreased insulin secretion using a static GSIS assay and observed that across both male and female mice there was a decrease in insulin secretion (Fig. S6E, F). Correspondingly, the *Rbfox2*-mutant mice also displayed decreased levels of circulating plasma insulin measured 15 min after intraperitoneal glucose injection, supporting defective insulin secretion in *Rbfox2*-mut mice (Figs. 4H, S6G). Furthermore, isolated β cells have decreased insulin exocytosis (Figs. 4I, S6H) despite the lack of changes in calcium current (Fig. S6I). Together this indicates that the *Rbfox2*-mut β cells have decreased granule docking at the plasma membrane resulting in reduced exocytosis, leading to overall decreases in glucose stimulated insulin secretion.

### RBFOX2 binds near splicing sensitive exons in β cells

To begin to dissect the molecular mechanism by which RBFOX2 is directly regulating its targets across the β cell transcriptome, we conducted two replicates of RBFOX2 and control IgG eCLIP-Seq experiments in MIN6 (Fig. S7A). As shown in Fig. S8, the two replicates are highly reproducible. Peak calling by *clipper*[43] and subsequent normalization by corresponding input samples identified 30,993 RBFOX2 peaks ($\log_2$Fold Change > 0, *p*-value < 0.05). Importantly, one of the most significantly enriched set of 6 nucleotides (6-mer) under the RBFOX2-eCLIP peaks was (U)GCAUG (Figs. S7B, S8A, B) and the mean peak width was 91nt (Fig. S7C, S8C, D). To further focus our analysis on top RBFOX2 targets, we selected for RBFOX2 peaks with a Fold Enrichment ≥ 2 and *p*-value < $10^{-3}$[43, 44]. This analysis identified 6,514 RBFOX2-eCLIP peaks within 2,494 genes (Fig. 5A). Of the eCLIP peaks found within genes, a majority of peaks were located in introns (86.4%) with additional peaks in exons (6.2%) and 3'UTRs (7.3%) but very few peaks in the 5'UTR (<0.1%) (Figs. S7D, S8E, F).

To determine the direct alternative splicing targets of RBFOX2, we overlapped the 2,494 bound genes with the previously characterized 606 alternatively spliced genes in the *Rbfox2*-KD MIN6 cells. 251 genes were significantly alternatively spliced and bound by RBFOX2. This accounted for 411 of the previously identified alternative splicing events, the majority being cassette/skipped exon (SE, 304) and mutually exclusive exon (MXE, 78) splicing events (Fig. 5A). In total, we observed 888 eCLIP peaks within genes that were aberrantly spliced when RBFOX2 is absent and the majority of these peaks were found within introns (79.2%, 703 peaks) (Fig. 5A, Supplementary Data 15).

Overall, there was an enrichment of RBFOX2 binding sites within ~500 nt of an alternative exon (Fig. 5B). More specifically, RBFOX2-eCLIP peaks were significantly enriched within 150nt immediately upstream of skipped alternative exons and downstream of included alternative exons when compared to exons that were not alternatively spliced, which are referred to as insensitive exons in SE splicing events (Fig. 5C, Supplementary Data 16). Furthermore, a similar trend was observed with the inclusion of both the first and second exons in MXE splicing events. RBFOX2 peaks were enriched within 150nt downstream of the first exon (Fig. 5D, Supplementary Data 16) or around the second exon (Fig. 5E, Supplementary Data 16), when each was independently included. RBPs can bind in introns, near intron-exon boundaries, to block or recruit splicing factors like the spliceosome

or other RBPs resulting in changes in alternative splicing. This data supports the model that RBFOX2 binds within introns adjacent to alternative exons to promote or inhibit exon splicing of both cassette/ skipped exons (SE) and mutually exclusive exons (MXE).

### Loss of RBFOX2 from β cells results in changes in cassette and mutually exclusive exon splicing of exocytosis and cytoskeletal related genes

We discovered that alternative splicing was often conserved at the same exon in RBFOX2-bound transcripts across *Rbfox2*-KD in MIN6 cell, *Rbfox2*-mut islet, and T2D islet datasets (Supplementary Data 17). To characterize the precise molecular functions of RBFOX2 that could be responsible for the *Rbfox2*-mut phenotype and were relevant to T2D, we focused on the intersection of differentially spliced genes across T2D islets, *Rbfox2*-mut islets, and *Rbfox2*-KD β cells that represented the top GO terms (Fig. 3F) and the transcripts that were both bound by RBFOX2 and whose splicing patterns changed in the presence or absence of RBFOX2 (Fig. S9A). Notably, transcripts involved in the "calcium regulated exocytosis of neurotransmitter", including SNARE complex components *Snap25*, Syt7, and *Stxbp1* were enriched in both datasets, and particularly corresponded to the observed phenotype. We determined the majority of these transcripts had significant changes in cassette/skipped exons (SE) and mutually exclusive exons (MXE). In addition to regulating the same exons, there was evidence that these exons were often mis-spliced in the same direction (inclusion vs skipping, represented by ΔPSI) (Fig. 6A). Furthermore, the majority of these alternative splicing events have one or more RBFOX2 eCLIP peaks near the RBFOX2-sensitive exon (Fig. 6B). Taken together, this data indicates that RBFOX2 may be regulating the normal splicing patterns of key exocytosis and cytoskeletal genes through a conserved mechanism that becomes disrupted in T2D.

To further investigate how RBFOX2 is directly affecting the splicing of insulin secretion pathway genes, we focused on a few key targets known to be important for insulin secretion, including *Snap25*, *Syt7*, and *Stxbp1*. SNAP25 is a component of the SNARE complex required to dock insulin granules to the plasma membrane[12] and is alternatively spliced across all three datasets in a mutually exclusive manner (Fig. 6A). *Snap25* has an evolutionarily duplicated exon 5[45,46], exon 5a and exon 5b, however, only one is expressed in each *Snap25* transcript. The sashimi plot in Fig. 6C shows that both exon 5a and exon 5b are expressed in β cells under normal conditions but that in the *Rbfox2*-KD there is decreased expression of exon 5b. This results in a shift in alternative splicing from a balance of both isoforms to an increase in isoforms containing the 5a exon as observed by the increase in change in percent spliced in (ΔPSI) in Fig. 6D and validated by qRT-PCR (Fig. 6E). Similar changes in ΔPSI were observed in the exocytosis-related genes *Syt7* and *Stxbp1* (Fig. S9B, C).

Consistent with the direct regulation of *Snap25* splicing by RBFOX2, the RBFOX2 eCLIP-Seq tracks identified three distinct peaks downstream of the second alternative exon, two of which contain the canonical GCAUG RBFOX2 binding consensus sequence (Fig. 6C, arrowheads). We independently validated that *Snap25* mRNA is a direct RBFOX2 target using RBP-immunoprecipitation (RIP) for RBFOX2 followed by qPCR for the *Snap25* RNA. This demonstrated a ~8-fold increase in *Snap25* mRNA in the RBFOX2-RIP compared to an IgG-RIP control confirming that RBFOX2 directly binds to the *Snap25* mRNA to regulate its splicing (Fig. 6F). Similarly, we observed RBFOX2 eCLIP peaks near alternative exons in *Syt7* and *Stxbp1* (Fig. S9B, C). This data indicates that these exocytosis-related genes are direct targets of RBFOX2-mediated splicing.

### Discussion

Tightly regulated insulin secretion is required to maintain blood glucose homeostasis. Disruption of this mechanism results in impaired insulin secretion and ultimately diabetes. Glucose stimulated insulin

## A. RBFOX2-eCLIP and alternative splicing overlap

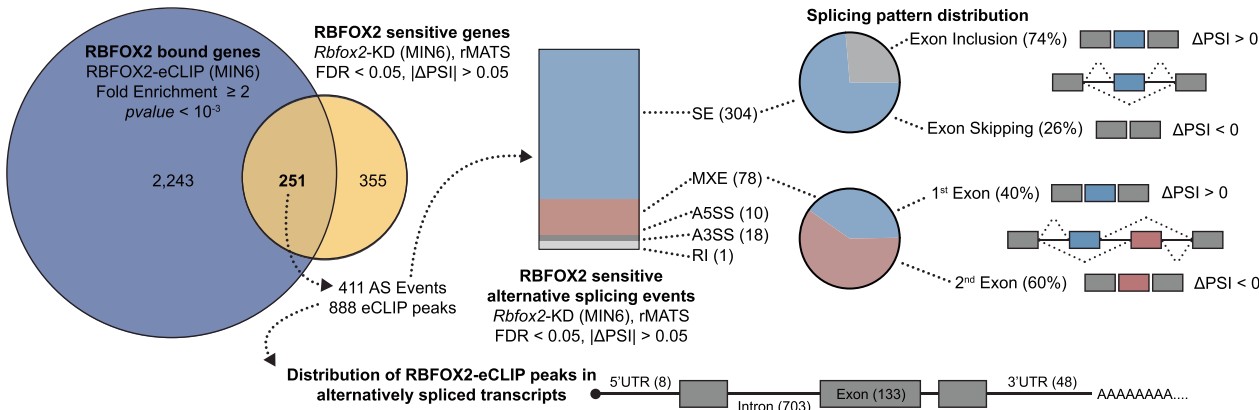

## B. RBFOX2-eCLIP enriched genes

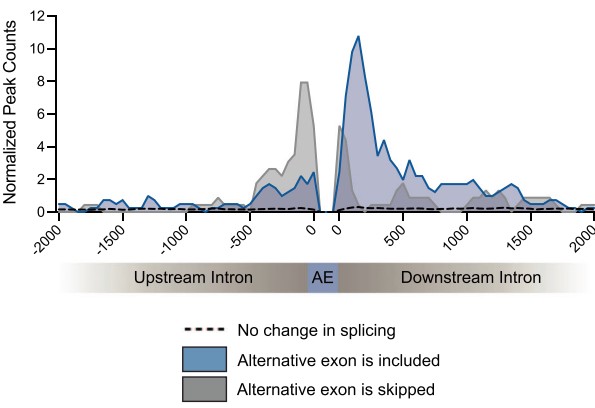

## C. RBFOX2-eCLIP peaks near SE events

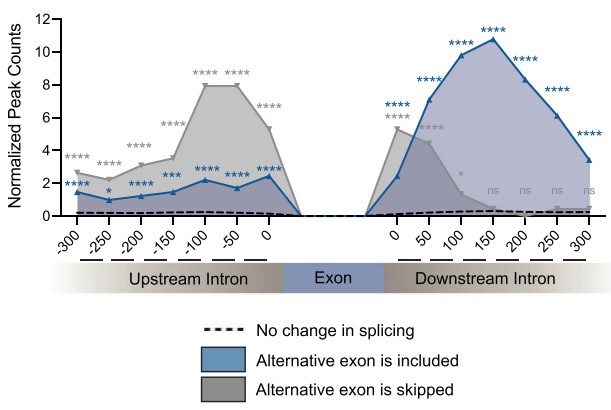

## D. RBFOX2-eCLIP peaks 1st exon in MXE

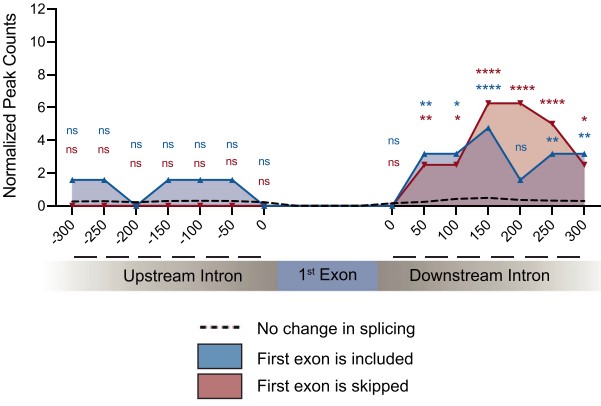

## E. RBFOX2-eCLIP peaks 2nd exon in MXE

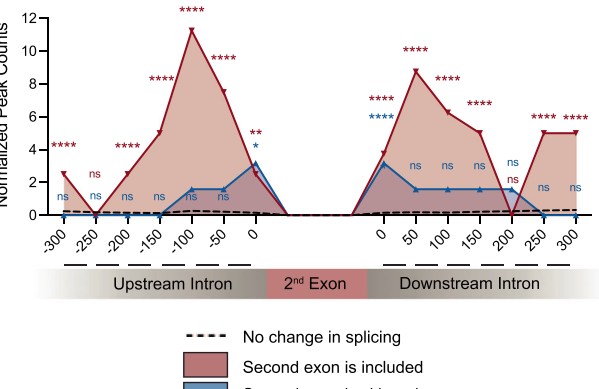

**Fig. 5 | Transcriptome wide assessment of RBFOX2 binding. A** Comparison of genes with one or more RBFOX2-eCLIP peaks identified by eCLIP-Seq with alternatively spliced transcripts from *Rbfox2*-KD in MIN6 cells identified by rMATs, cassette/skipped exon (SE), mutually exclusive exon (MXE), alternative 5' start site (A5SS), alternative 3' splice site (A3SS), or retained intron (RI) to identify RBFOX2 direct splicing targets, statistical significance was determined by rMATS for splicing analysis and *Clipper* for eCLIP analysis. **B** Distribution of RBFOX2 eCLIP-peaks relative to alternatively spliced included cassette exons (blue), skipped cassette exons (gray), or insensitive exons where splicing is not significantly changed between *Rbfox2*-KD and control (dashed line) 2000nt up and downstream of alternative exons. **C** Distribution of RBFOX2 eCLIP-peaks relative to alternatively spliced included cassette exons (blue), skipped cassette exons (gray), or insensitive

exons where splicing is not significantly changed between *Rbfox2*-KD and control (dashed line) 300nt up and downstream of alternative exons. **D** Distribution of RBFOX2 eCLIP-peaks relative to alternatively spliced included 1st exon in mutually exclusive exons (blue), skipped 1st exon in mutually exclusive exons (red), or insensitive exons (dashed line). **E** Distribution of RBFOX2 eCLIP-peaks relative to alternatively spliced included 2nd exon in mutually exclusive exons (red), skipped 2nd exon in mutually exclusive exons (blue), or insensitive exons (dashed line). **C–E** Bootstrapping of eCLIP peaks at RBFOX2 insensitive exons was used to identify the probability and distribution of RBFOX2 binding events (peaks) and significant enrichment was calculated using a Poisson distribution, (ns FDR > 0.05, *FDR ≤ 0.05, **FDR ≤ 0.01, ***FDR ≤ 0.001, ****FDR < 0.0001).

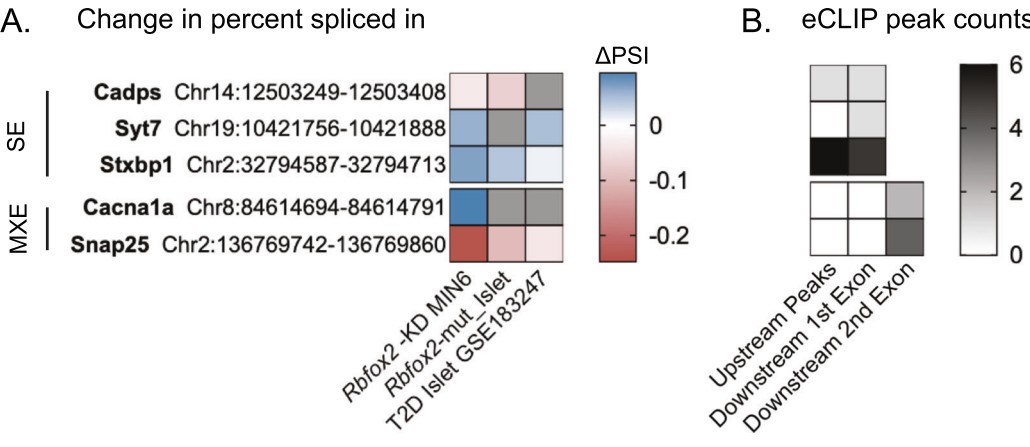

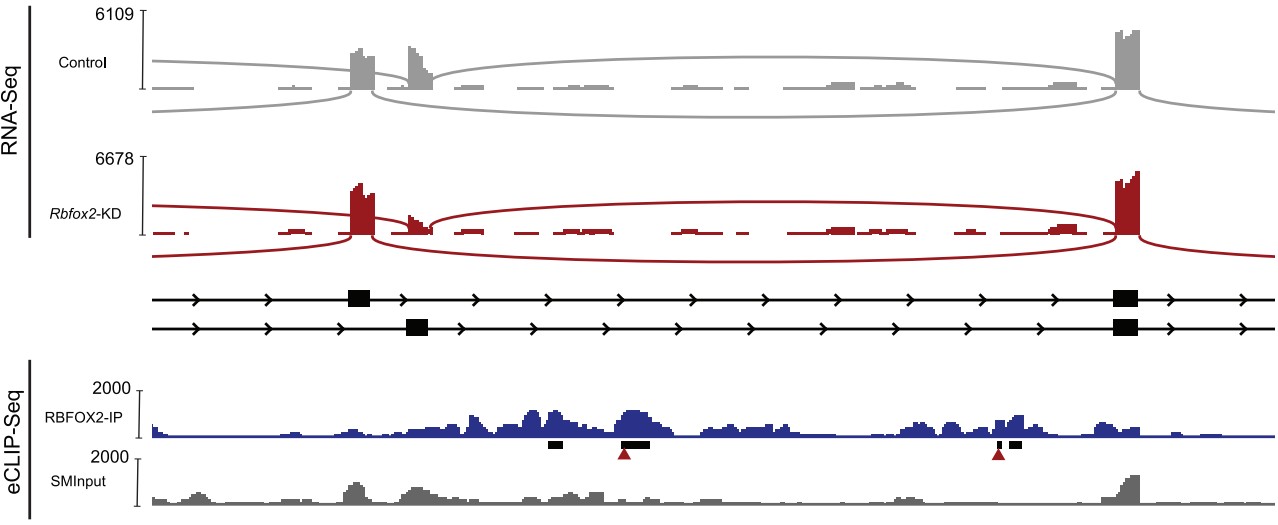

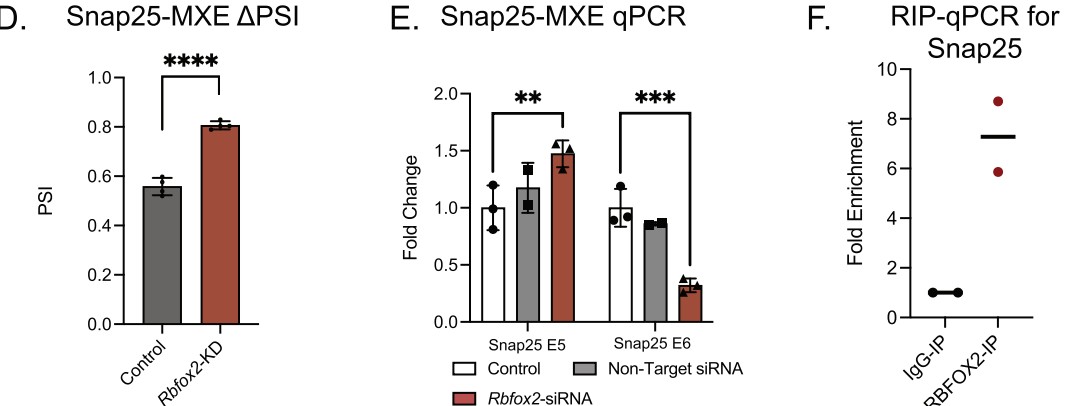

**Fig. 6 | RBFOX2 binding and regulation of alternative splicing in genes involved in insulin secretion. A** Heatmap of shared alternatively spliced exons (cassette/skipped exon, SE and mutually exclusive exon, MXE) across *Rbfox2*-KD in MIN6, *Rbfox2*-mut islets, and T2D islets (*ob/ob*, NZO) color indicates △PSI from rMATS, gray boxes indicate no significant splicing identified in specific exon. **B** eCLIP peak count in upstream or downstream introns in MIN6 cells. **C** Sashimi plot for *Snap25* MXE with eCLIP peaks, RBFOX2 eCLIP-Seq (blue), *Clipper* identified significant peaks (black bars), GCAUG sequence (red triangles), SMI eCLIP-Seq (gray). **D** *Snap25* MXE percent spliced in (△PSI), *n* = 4 per condition,

FDR < 0.00001 significance determined by rMATs. **E** qRT-PCR validation of *Snap25* MXE alternative splicing event in *Rbfox2*-KD vs control, *n* = 4, two-way ANOVA with multiple comparisons (*Snap25*-5a: Non-Treated vs. Non-Target padj = 0.3601, Non-Treated vs. *Rbfox2*-KD padj = 0.005, *Snap25-5b*: Non-Treated vs. Non-Target padj = 0.5061, Non-Treated vs. *Rbfox2*-KD padj = 0.0004). **F** RIP-qPCR validation of RBFOX2 binding to *Snap25* in MIN6 (*n* = 2). (Data are represented as mean values with SD, ns *p* > 0.05, **p* ≤ 0.05, ***p* ≤ 0.01, ****p* ≤ 0.001, *****p* < 0.00001).

secretion is regulated at many levels, including insulin expression, processing and packaging, zinc levels, and modulation of ion channels. We have uncovered a novel mechanism of insulin secretion regulation through alternative splicing of exocytosis-related genes. In this study, we show that RBFOX2 is critical in the fine-tuning of alternative splicing across several genes that regulate insulin granule organization and exocytosis. Loss of RBFOX2 from the pancreas results in a shift in splicing and isoform expression of *Snap25*, *Stxbp1* (*Munc18-1*), and *Syt7*, among other genes, causing a disruption in insulin granule docking and impaired insulin secretion (Fig. 7). Notably, the same mis-splicing events are also observed in the pathogenesis of T2D and may explain much of the disruption in first-phase insulin secretion observed in pre-diabetes and diabetes[6,7]. Our in-depth phenotypic analysis of *Rbfox2*-mut mice showed that, despite a lack of changes in β cell number or insulin content, there are fewer docked insulin granules and a significant decrease in first-phase insulin secretion (Fig. 7). Similar defects in insulin granule docking have been shown to be responsible for the rapid response to changes in blood glucose in first-phase insulin secretion and consistent with the diabetic phenotype[5–7].

Our analysis indicates that RBFOX2 fine-tunes insulin granule organization, through its alternatively spliced targets, ultimately impacting insulin secretion. RBFOX2 splicing targets are enriched for genes involved in insulin granule trafficking and exocytosis (Fig. 3G). Interestingly, these shifts in alternative splicing that result from loss of RBFOX2 from both the mouse pancreas and β cells specifically are mimicked in the mouse model of T2D[21]. Many of these splicing-sensitive genes, including *Snap25*, are known to have alternative isoforms with differential expression and function[46–49]. Research in neurons has highlighted the functional differences and differential binding partners of the *Snap25a* and *Snap25b* isoforms[45,46,48,49]. For example, the SNAP25b isoform has been shown to differentially interact with STXBP1[48], and SNAP25b is required for synapse maturation[45,49]. One study indicated that deficiencies of SNAP25b in β cells result in increased insulin secretion[50]. In the *Rbfox2*-mut mice, we see a similar deficiency in *Snap25b*, but in combination with disruption of other SNARE complex components including the SNAP25 binding partner STXBP1. Here there is a significant decrease in insulin secretion, consistent with previous studies that showed deficiencies in multiple SNARE complex components[15]. Furthermore, a shift in insulin granule

distribution along the plasma membrane, similar to what is observed in the *Rbfox2*-mut mice, is a predominant limiting factor resulting in decreased insulin secretion[5]. We hypothesize that the splicing disruption of multiple SNARE complex and related proteins (*Snap25*, *Syt7*, *Stxbp1*, and *Cadps*) combined with granule trafficking-related genes (*Mapt*, *Dcnt1*, *Slain2*, *Clip1*, etc.) results in the observed disruption of granule docking and decrease in insulin secretion.

Interestingly, although we observe alterations in the splicing of *Cacna1a*, the main calcium channel for exocytosis in mouse β cells, we do not see corresponding defects in calcium currents in the *Rbfox2* mutant islets. Because the *Cacna1a* gene contains 47 exons, many of which are subject to alternative splicing, the number of CACNA1A splice isoforms is estimated to be in the order of thousands, many of which are functional. Therefore, it is possible that the altered protein isoforms of CACNA1 resulting from the loss of *Rbfox2* do not affect protein function or that the relative ratio of isoforms is not sufficiently skewed to cause a phenotype. Future studies will explore the potential implications of alternative splice variants in the *Cacna1a* gene.

Although our study suggests that many of the aberrant splicing events observed in T1D and T2D are due in part to the loss of RBFOX2 function from pancreatic β cells, how RBFOX2 itself becomes dysregulated under the stress conditions associated with chronic high glucose or autoimmunity will be difficult to tease apart. RBFOX2 is regulated both transcriptionally and post-transcriptionally, including at the level of splicing. Moreover, RBFOX proteins have been shown to be autoregulatory[51], which further complicates this type of analysis. However, with the advent of whole genome sequencing, improved read depth, and long-read sequencing of human islet samples, RBFOX2 will likely emerge as a diabetes risk gene RBFOX2 function has also been characterized across many cell types and systems, including β cells[18], neurons[37,52,53], muscle cells[31,34–36], stem cells[29], and in cancers[54]. Interestingly, Juan-Mateu et al. investigated the role of RBFOX1 and RBFOX2 in rat and human immortalized β cell lines and suggested a role for RBFOX proteins in insulin secretion[18]. However, this study also concluded that mis-splicing of gelsolin, *Gsn*, by RBFOX1, caused an increased insulin secretion phenotype[18]. Our investigation of RBFOX2 targets, which shares a binding consensus sequence and many binding targets with RBFOX1[27], using RNA-Seq and eCLIP-Seq techniques failed to identify gelsolin (*Gsn*) as a splicing target of RBFOX2, despite the

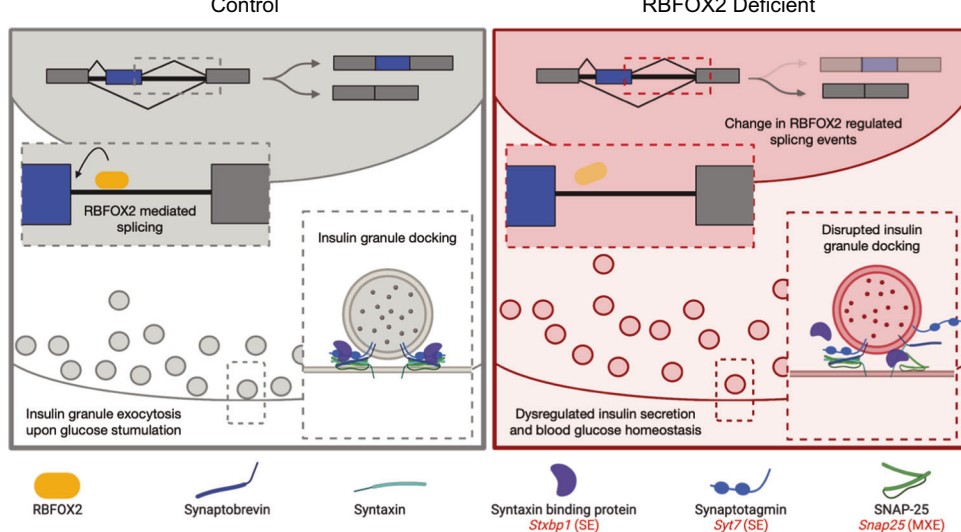

**Fig. 7 | Model of RBFOX2 mediated regulation of insulin granule secretion in β cells.** RBFOX2 (yellow) binds downstream of alternative exons to promote exon inclusion in key genes involved in insulin granule docking and exocytosis (including *Stxbp1, Syt7*, and *Snap25*). Splicing of these genes allows for proper docking and exocytosis of insulin granules. Loss of RBFOX2, results in mis-splicing of insulin granule docking and exocytosis genes. Causing a decrease in the docked pool of insulin granules and corresponding decrease in insulin secretion (created with BioRender.com).

presence of an RBFOX consensus sequence. Furthermore, we were unable to detect the expression of *Rbfox1* in human or mouse β cells (Figs. S1D, S2A). We are uncertain how to explain these discrepancies, but our studies ultimately agree that RBFOX2 is an important regulator of insulin secretion. Additionally, our observations of alterations in splicing and RBFOX2 binding in genes involved in the regulation of cytoskeleton organization, vesicle-mediated transport, and calcium signaling have also been observed in neurons suggesting a conservation of function[18,52].

Consistent with studies in other systems, we show that RBFOX2 can act to activate and repress splice sites depending in part on binding localization relative to the alternative exon[52,53,55]. Our data supports a conserved mechanism for RBFOX2 regulation of alternative exons in β cells. In RBFOX2 sensitive cassette/skipped (SE) or mutually exclusive exons (MXE), exon inclusion is driven by RBFOX2 binding is enriched downstream of the alternative exon. It is important to note that RBFOX2 is likely not acting alone in this system[56] and that interactions with other RBPs may further elucidate the model of RBFOX2 regulation in β cells. This could in part explain incidences where RBFOX2 regulation does not follow the predicted pattern of downstream binding and inclusion in our datasets. Furthermore, our splicing analysis captures a subset of the RBFOX2-bound genes indicating it may have additional roles. Recent studies have also suggested alternative roles for RBOFX2 in the 3'UTR to regulate transcript stability[54] and poly-adenylation[57] to help regulate expression levels of genes in stress granules[32]. Future studies would focus on investigating the complex network and coordination of RBPs in RBFOX2-mediated regulatory events.

In this study, we have begun to resolve the molecular mechanism for the fine-tuning of insulin secretion through RBFOX2-mediated alternative splicing. There is emerging interest in the therapeutic deployment of antisense oligonucleotides (ASOs) as a mechanism to rectify aberrant splicing events in specific RNA transcripts[58]; however, it is clear from studies in both human diabetic islets and mouse models of diabetes, that thousands of splicing events are disrupted. This study demonstrates that misexpression of RBFOX2 causes significant changes in alternative splicing that leads to dysfunctional β cells, and RBFOX2 binding is sensitive to RBFOX2 concentration[26], we hypothesize that restoring normal levels of RBFOX2 expression may resolve aberrant splicing leading to disrupted insulin secretion that is observed in T2D.

Despite the limited availability, variability, and relatively low quality of human datasets we observe both trending differential expression of *RBFOX2* and enrichment of the RBFOX2 consensus binding sequence downstream of included exons. Increasing the sequencing quality and depth in future human analyses will vastly improve the robustness of the datasets and the ability to confidently identify changes in alternative splicing[59–61].

We have taken a multi-model approach to investigate the role of RBFOX2 in the mouse pancreas. By paring in vivo experiments in *Rbfox2*-mut mice with *Rbfox2*-KD in MIN6 cells, we leverage the benefits of each system and confirm cell-type-specific contributions and splicing changes. Decreased expression of *Rbfox2* is observed in islets that contain multiple endocrine cell types, including the β cell. Our use of the Pdx1:CRE allele[42], allows us to model similar decreased *Rbfox2* expression across cells of the islet as is observed in T2D. To model what changes in splicing events are occurring specifically in the β cell, we shifted to the MIN6 cell line. The MIN6 cells provide a robust model to investigate RBFOX2 β cell-specific binding. The intersection of these datasets identifies the contributions of β cell-specific splicing changes and RBFOX2 targets relevant to the phenotype loss across endocrine cell types observed in *Rbfox2*-mut islets and T2D islets. We hypothesize that similar studies in α-TC[62] would identify α cell-specific and endocrine-conserved targets for RBFOX2. For example, much of the exocytosis-related machinery we identified as RBFOX2 targets in the β

cell, including components of the SNARE complex[63], are involved in glucagon secretion in α cells, which is also disrupted in diabetes[64]. Perhaps targeting decreased *Rbfox2* expression across the islet cell types will restore not only robust insulin secretion from β cells but restore the tightly regulated multi-dimensional process of maintaining blood glucose homeostasis.

## Methods

### Experimental model and subject details

**Cell line and cell culture.** Mouse insulinoma cells (MIN6)[65] are not commercially available and were obtained from the Yamamura lab. The cells were maintained using standard techniques, 37 °C in 5% $CO_2$ and passaged every 7 days. Cells were cultured in Dulbecco's modified eagle's medium (DMEM) with 10% FBS and 1% Penicillin-Streptomycin.

**Mouse studies.** In this study, we generated *Rbfox2* conditional mutant mice by combining a single *Pdx1:CRE* allele (B6.FVB-Tg(Pdx1-cre) 6Tuv)[42] with one or two *Rbfox2*^flox alleles (B6.129S2-*Rbfox2*^*tm1.1Dblk*/J)[37]. Unless otherwise noted, all experiments in this study were conducted on 8-week-old adult male and female mice. The mice were maintained under the University of Colorado Institutional Animal Care and Use Committee (IACUC) approved protocol #00045. Mice were housed by sex with up to 5 littermates per cage in a 22 °C room with 12-h light/ dark cycle and unlimited access to food (Inotivco 2920X) and water. Cages were cleaned every 2 weeks and mice were regularly monitored for injury and infection. Euthanasia was performed with $CO_2$ and cervical dislocation. Genotyping primers are listed in Supplementary Data 18.

**RNA-binding protein identification.** To identify RNA-binding proteins (RBP) relevant to diabetes pathology we evaluated top RBP expression level changes in diabetic mouse islets compared to controls. We established a pipeline to reanalyze bulk-RNA-sequencing data from GSE183247[21] to identify top differentially expressed RBPs. Data was obtained from GEO database and aligned to the mouse genome (mm10) using STAR 2.7.9a[66] and manipulated using samtools[67]. Reads were counted using featureCounts (subread v1.6.2)[68] and subsequently compared using DESeq2 (v1.34.0)[69]. RNA splicing related genes were identified using the RNA splicing gene ontology list (GO:0008380). The most highly expressed and differentially expressed in this category were filtered for a base mean expression of >1500 in either the NZO or Db/Db groups and a significant differential expression as identified by DESeq2 above.

To identify enriched 5-mers near alternative exons we first identified significantly changed cassette exons (CE), also referred to as skipped exons (SE), using the splicing analysis package rMATS (v4.0.2). We then investigated intron sequences 200nt up and downstream of the alternative exon for each possible 5-mer using a sliding window and identified enrichment by comparing to 5-mers identified in cassette exons identified but not significantly changed in the dataset. A similar analysis was performed on intron sequences 200nt flanking alternative SE in human alternatively spliced genes from healthy and type-II diabetic donors identified by rMATS v.4.0.2 (GSE164416) Original code for the 5-mer analysis will be deposited at https://github.com/ CUAnschutzBDC.

**Immunohistochemistry and insulin area quantification.** Pancreata were collected from 8-week-old male mice and fixed in 4% paraformaldehyde (PFA) for 4 h at 4 °C. The fixed tissue was then prepared for cryo-preservation by acclimation in 30% sucrose overnight at 4 °C before embedding in Optimum Cutting Temperature (OCT) and frozen at −80 °C. Frozen blocks were then cut into 10 μm sections on a Microm HM 525 cryostat. Slides were then labeled with antibodies to identify cell types by hormone expression. Tissue sections were first permeabilized in PBS with 0.01% Triton-X (PBS-T) and blocked with 2%

Normal Donkey Serum (NDS) for a minimum of 30 min. Sections were then incubated with a primary antibody diluted in 2% NDS overnight at 4 °C. After three washes with PBS-T slides were incubated in secondary antibody and DAPI (1:1000, Fisher Scientific, D1306) for 2 h at RT. After three additional washes in PBS-T, slides were mounted with VECTA-SHIELD Hardset Antifade Mounting Medium (Vector Laboratories, H-1400). All primary and secondary antibodies are listed in Supplementary Data 19.

High magnification images were obtained using a Zeiss Confocal LSM800 microscope and ImageJ software. Lower magnification images used for insulin and c-peptide 1 area quantification were obtained using a Leica DM5500B microscope. Images used to quantify insulin area were labeled with anti-insulin, anti-somatostatin, and DAPI. In ImageJ thresholding of the insulin color channel identified the insulin positive area within the islet. Total islet area was outlined, identified by DAPI+ cells and used to normalize image quantification.

**Blood glucose measurements and glucose tolerance test.** Blood samples collected from the tail vein were used to measure blood glucose levels using Contour 7151H blood-glucose monitor with Contour 7097 C blood glucose strips. For both intraperitoneal glucose tolerance test (IP-GTT) and oral glucose tolerance tests (O-GTT), mice were fasted overnight for ~12 h. An initial blood glucose reading was taken to measure fasting blood glucose. Mice were then administered 2 mg D-glucose per gram body weight either via intraperitoneal injection (IP) or via oral gavage for the IP-GTT and O-GTT respectively. Blood glucose measurements were taken at 15, 30, 60, 90, and 120-min following glucose administration. A minimum of three biological replicates were used to compare the area under the curve (AUC) for each condition.

**Plasma insulin.** Glucose was administered by IP injection at 2 mg per gram body weight. Whole blood was collected 15 min following injection and plasma insulin content measured by the Mouse ULTRA Sensitive Insulin ELISA (Alpco) and read on Biotek plate reader.

**Islet Isolation.** Islet isolations were performed by the Islet Core Facility at the Barbara Davis Center at the University of Colorado, Anschutz Medical Campus. Islets were isolated by perfusion of 2.5 mL of cold CIzyme solution through the common bile duct into the pancreas. Pancreas was excised and incubated at 37 °C until exocrine and endocrine tissue became separated. Wash with cold HBSS and purified via a density gradient (Lympholyte 1.1) and washed remaining endocrine tissue with HBSS and 10% fetal calf serum and finally handpicked. Islets were cultured overnight at 37 °C in 5% $CO_2$ in RPMI1640 with 10% FBS and 1% Penicillin-Streptomycin.

**Static glucose stimulated insulin secretion.** Isolated and rested islets were separated in to triplicate pools of 10 islets for each sample and condition. Islets were initially equilibrated for 1 h in low glucose media (Krebs buffer, 0.1% BSA, and 2 mM glucose), followed by a 30 min treatment in low glucose, high glucose (Krebs buffer, 0.1% BSA, and 20 mM glucose), or low glucose with potassium chloride (Krebs buffer, 0.1% BSA, 2 mM glucose, and 20 mM KCl). Secreted insulin was collected from supernatant and remaining islets were lysed with 2% Triton-X for insulin extraction to obtain insulin content. Insulin content is measured by ELISA (Alpco) and read on Biotek plate reader.

**Dynamic glucose stimulated insulin secretion.** Mouse insulin perifusion protocol has been detailed previously and published on protocols.io[70]. Briefly, 35 mouse islets are perfused with glucose and KCL as indicated. Samples are collected at 2–5 min intervals.

**Patch-clamp electrophysiology studies.** Single-cell patch-clamp studies were carried out as we described previously[71]. Control and

RBFOX2 KD mouse islets were dissociated to single cells and cultured for 1–2 days. Prior to whole-cell patch clamping, media was changed to a bath solution containing (in mM): 118 NaCl, 20 Tetraethylammonium-Cl, 5.6 KCl, 1.2 $MgCl_2$, 2.6 $CaCl_2$, 5 HEPES, and 5 glucose (pH 7.4 with NaOH) in a heated chamber (37 °C). Fire polished thin wall borosilicate pipettes were coated with Sylgard (3-5mOhm) and filled with the following pipette solution (in mM): 125 Cs-glutamate, 10 CsCl, 10 NaCl, 1 $MgCl_2$, 0.05 EGTA, 5 HEPES, 0.1 cAMP, and 3 MgATP (pH 7.15 with CsOH). Electrophysiological data were recorded using a HEKA EPC10 amplifier and PatchMaster Software (HEKA Instruments Inc, Lambrecht/Pfalz, Germany) within 5 min of break-in. FitMaster (HEKA Instruments Inc) were used for data analysis. To measure exocytotic response as increases in cell surface area (capacitance) cells were held at −70 mV and subjected to a series of ten 500 ms depolarizations to 0 mV. Total exocytotic responses were taken at the difference between cell capacitance before depolarization and after the 10th pulse (in fF) and normalized to initial cell size (in pF).

**Electron microscopy and image quantification.** Isolated islets from three replicate *Rbfox2*-mut and control mice were transferred to the University of Colorado Boulder Electron Microscopy Core where the samples were prepped. Isolated islets were high pressure frozen on a Wohlwend Compact02 HPF (cryo-protectant was 10% dextran/150 mM mannitol in culture media), followed by freeze-substitution (low temperature fixation) in 2% osmium tetroxide/0.2% uranyl acetate in anhydrous acetone. Resin (Embed812/Araldite) infiltration occurred at RT over 4 days, and blocks containing samples were then polymerized at 65 °C for 48 h. Using a Leica UCT ultramicrotome, thin sections (60-80 nm) were cut and collected on Formavar-coated TEM slot grids. Samples were then post stained with 2% aqueous uranyl acetate and Reynold's lead citrate before imaging on a Tecnai T12 Spirit TEM, operating at 100 kv, with an AMT CCD digital camera. Images were taken 30,000x.

Image quantification was completed in ImageJ on a 1500 nm x 3500 nm region of each image containing the interface of two adjacent β cells. Area of each cell was quantified as well as mean distance from the edge of the frame to the membrane to ensure comparable datasets between genotypes. The diameter and perpendicular distance to the membrane were measured for each insulin granule in the frame. Overall, 75 mutant and 71 control β cells were quantified across three biological replicates for each genotype, for a total of 700 and 657 granules respectively.

**RNA interference.** MIN6 cells were grown to 70% confluence in 6 well plates then transfected with 150 pmol of 4 pooled siRNAs targeting *Rbfox2* exons or a pool of 4 scramble using RNAi MAX (Thermo) per manufacturer instructions. Cells were harvested after 48 h and *Rbfox2* knockdown was assessed by western blot and qRT-PCR. Sequences for the siRNAs are listed in Supplementary Data 18.

**Western blotting.** Cell lysates from *Rbfox2* knockdown and control MIN6 cells were collected. Approximately 50 µg of each sample was loaded in to a 4-20% Bis-Tris polyacrylamide gel (Invitrogen) and then proteins were transferred to PVDF membrane. The membrane was then blocked with 5% milk in TBST for at least 1 h at RT before incubating with primary antibody for RBFOX2 (Bethyl, A300-864A) and the loading control β-catenin (Abcam, 32572) overnight at 4 °C. Blot was washed in TBST, incubated in secondary antibody (Rb anti-HRP, Abcam, 205718), washed in TBST, and developed with Western Lightning chemiluminescence kit (GE Biosciences) on Azure3000 imager.

**RNA extraction and qRT-PCR.** Total RNA was extracted from isolated islets using the RNEasy Plus Micro Kit (Qiagen) or MIN6 cells using the RNEasy Plus Mini Kit (Qiagen). The RNA was then quantified and checked for quality by Nanodrop 2000 (Thermo Fisher) or Qubit

(Thermo Fisher, Q33226). A minimum of 200 ng of total RNA was then used to synthesize cDNA with iScript cDNA synthesis kit (Biorad) and the resulting cDNA was diluted to 4 ng/μL. A total of 16 ng/μl were used in each qRT-PCR reaction along with SsoAdvanced Universal Probes Supermix (Biorad) with Taqman probes or Syber green (Invitrogen) with primers (IDT), probes and primers are listed in Supplementary Data 18. Samples were run on a Bio-Rad CFX96 real time PCR detection system and expression levels were normalized to *Actb*. Sample comparison was determined using the $2^{-\Delta\Delta Ct}$ method.

**RNA sequencing and analysis.** Total RNA from mouse islets or treated MIN6 cells was extracted as described above and tested for quality using an Agilent 2100 Bioanalyzer. Only samples with RIN > 8.0 were used in RNA sequencing experiments. Libraries were prepared using Universal Plus mRNA-Seq library preparation kit with NuQuant sequenced by the Genomics Core at the University of Colorado Anschutz Medical Campus using the NovaSEQ 6000 for paired end sequencing (2x150) from PolyA selected total RNA. Reads were checked for quality using FastQC (v0.11.9) before and after adapters were removed with Cutadapt[72]. Reads were aligned to the mm10 genome using STAR aligner (2.7.9a) and manipulated using samtools[67].

Differential expression was determined by counting reads using featureCounts (subread 1.6.2)[68] and subsequently compared using DESeq2 (v1.34.0)[69]. Resulting differentially expressed genes were filtered for adjusted *p*-value < 0.05 and a minimum of 20 base mean reads in one of the two conditions. Alternative splicing was analyzed by rMATS (v4.0.2)[73] and resulting splicing events were filtered for 20 reads across samples in one of the two conditions, FDR < 0.05, and |ΔPSI| > 0.01 or |ΔPSI| > 0.05. Splicing insensitive events were identified to have an FDR > 0.05. Pipelines and analysis tools are listed in the Supplementary Data 20 and are available at https://github.com/CUAnschutzBDC.

**RIP-qPCR.** RNA-binding protein immunoprecipitation and qPCR was performed on 100 μL of MIN6 cell lysate. MIN6 were cultured in 15 cm dishes to 70% confluence and washed with cold PBS before collection. Cell pellet was resuspended in an equal volume of RIP Lysis buffer from the Magna RIP kit (Millipore) to lyse the cells. Cell lysates were processed following the Magna RIP kit (Millipore) protocol detailed in the Supplementary Data 19. Immunoprecipitation was performed using the anti-RBFOX2 rabbit polyclonal antibody (Bethyl, A300-864A) and anti-IgG rabbit antibody (Chemicon, PG64 25050057). Precipitated RNA was quantified using Nanodrop 2000 (Thermo Fisher) or Qubit (Thermo Fisher, Q33226) and converted to cDNA using the iScript cDNA Synthesis Kit (BioRad) and the resulting cDNA was diluted to 4 ng/μL. A total of 16 ng/μl were used in each qRT-PCR reaction along with SsoAdvanced Universal Probes Supermix (Biorad) with Taqman probes to quantify gene target enrichment, list of probes is available in Supplementary Data 18. Sample comparison was determined against the input using the $2^{-\Delta\Delta Ct}$ method and fold enrichment was compared relative to the IgG control.

**eCLIP sequencing and analysis.** eCLIP sequencing was performed on samples collected from MIN6 cells cultured in 15 cm dishes at 70% confluence. Cells were UV crosslinked twice on ice at 150 mJ/cm² using Stratagene UV Stratalinker 2400 and processed using the eCLIPse Bioinnovations RBP-eCLIP kit and protocol (RBP-eCLIP Protocol v1.01 R, #ECK001) detailed in the Supplementary Data 19. Immunoprecipitation was performed using the anti-RBFOX2 rabbit polyclonal antibody (Bethyl, A300-864A) and anti-IgG rabbit antibody (Chemicon, PG64 25050057). Libraries were sequenced by the Genomics Core at the University of Colorado Anschutz Medical Campus using the NovaSEQ 6000 for paired end sequencing (2x150). Library complexity was determined using preseq (v3.1.1) before analysis. Resulting sequences were analyzed using an adapted pipeline following the

original eCLIP analysis pipeline[43]. Significant eCLIP peaks (RBFOX2 binding events) were identified by Clipper. Position dependent binding was determined for RBFOX2-sensitive cassette exon and mutually exclusive exon splicing events and compared to insensitive or exons that were not significantly alternatively spliced in the *Rbfox2*-KD dataset. Bootstrapping of position dependent binding around insensitive exons was used to identify probability of a peak near an exon and is well represented by a Poisson distribution. Pipelines, additional description, and analysis tools are listed in the Supplementary Data 20 and is available at https://github.com/CUAnschutzBDC.

## Quantification and statistical analysis

Plots were generated in RStudio v4.1.3 and GraphPad Prism 9. Statistical analysis was preformed GraphPad Prism 9, all replicates, statistical tests, and significance are noted in Figure legends. A single outlier was identified and removed from the plasma insulin measurements the using the exclusion criteria ROUT ($Q = 0.5\%$) method.

### Reporting summary

Further information on research design is available in the Nature Portfolio Reporting Summary linked to this article.

## Data availability

This paper re-analyzes existing publicly available data located at GEO - GSE183247 originally published by Schurmann et al.[21]; GSE164416 originally published by Wigger et al.[40]; and PANC-DB [https://hpap.pmacs.upenn.edu/explore/download?matrix] originally published by Kaestner et al.[41]. The accession numbers for the data generated in this study are available through the SuperSeries accession number GSE221277, or individually at GSE221274 (Rbfox2-KD RNA-Seq MIN6), GSE221275 (Rbfox2-mut RNA-Seq Mouse Islet), and GSE221276 (RBFOX2 eCLIP-Seq MIN6). Source data are provided with this paper.

## Code availability

This paper utilized both established packages and original code. Original code and pipelines will be deposited at https://github.com/CUAnschutzBDC (https://doi.org/10.5281/zenodo.8338787, https://doi.org/10.5281/zenodo.8335301, https://doi.org/10.5281/zenodo.8335366) and will be publicly available. All established packages are listed in the Supplementary Data 20. Custom code and analysis protocols for this paper is available at https://github.com/CUAnschutzBDC/RBFOX2_project.

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

## Acknowledgements

We acknowledge members of the Sussel lab and in particular Dr. David Lorberbaum for critical reading of the manuscript. Islet isolation of islets was performed by Scott Beard in the Islet Isolation Core Facility at the Barbara Davis Center. Electron microscopy sample preparation and imaging was done at the University of Colorado, Boulder EM Services Core Facility in the MCDB Department, with the technical assistance of facility staff including Garry Morgan and Eileen O'Toole. Sequencing was performed by the University of Colorado Anschutz Medical Campus Genomics Shared Resource. The authors also acknowledge assistance from the University of Colorado Diabetes Research Cores funded by P30 DK116073. This work was supported by NIH Grants NIDDK U01 DK127505, R01 DK125360, R01 DK82590, P30 DK116073 (LS), F31 DK126320 (NDM), F31 DK122634 (AT), Diabetes Research Connection (YKK), and the University of Colorado RNA Biology Initiative (NDM, KLW). Additional support for the MacDonald lab included a Canadian Institutes of Health Research Foundation Grant (FS 148451) to P.E.M. and an NSERC-CREATE fellowship from the Canadian Islet Research and Training Network (CIRTN) to X.L. PEM holds the Canada Research Chair (Tier 1) in Islet Biology.

## Author contributions

N.D.M. performed the majority of the experiments, data analysis, and manuscript preparation. K.L.W. assisted with all aspects of the data analysis. Y.-K.K. and A.T. assisted in metabolic assays. Under the guidance of P.E.M., X.L. performed the patch-clamp electrophysiology assays, A.F.S. performed the dynamic secretion assays. L.S. conceived of the study and guided the project, assisted with interpreting data, and wrote the manuscript with N.D.M.

## Competing interests

The authors declare no competing interests.
