## [Peer review file · Nature Communications]

Modulation of insulin secretion by RBFOX2-mediated alternative splicingREVIEWER COMMENTS

Reviewer #1 (Remarks to the Author):

The regulation and function of alternative splicing in pancreatic islets and its role in diabetes is an exciting topic that remains relatively understudied. In this manuscript, Moss, Sussel and colleagues provide multiple evidences that the RNA-binding protein RBFOX2 plays an important role in pancreatic beta cell function by regulating the alternative splicing of genes involved in insulin granule docking and exocytosis. Using a conditional islet-specific Rbfox2 mutant mice, the authors show that lack of Rbfox2 causes reduced glucose-stimulated insulin secretion and glucose intolerance. Combining RNAseq analysis of Rbfox2 mutant mouse islets and eCLIP experiments in the MIN6 beta cell line, the authors identified hundreds of alternative splicing events directly regulated by Rbfox2 in genes relevant for insulin release, particularly in genes involved in granule docking and exocytosis. In line with this, using electron microscopy the authors show that Rbfox2 depletion causes alterations in insulin granule organization and a reduction in the number of docked granules to the plasma membrane. Interestingly, the authors also provide evidences that Rbfox2 is mis-regulated in diabetic conditions, correlating with splicing mis-regulation in T2D islets, and propose the exciting hypothesis that these splicing changes may contribute to the defects in the first phase of insulin secretion observed in T2D patients.

Overall, this is a novel and very interesting work that demonstrates a cellular and physiological role of Rbfox2-regulated splicing in pancreatic beta cell function and glucose homeostasis, expanding the current knowledge about the roles of alternative splicing in islets.

I have only minor comments:

1. The authors show in Figure 1 that RBFOX2 is down-regulated in diabetic conditions and that this correlates with alternative splicing changes observed in T2D. While the data is convincing in mice, is somehow weaker in humans probably due to the low number of samples analysed. Data from a larger number of donors (both for bulk and single cell RNAseq) is publicly available and would enable them to increase the power and robustness of their analysis. Additionally, it would be interesting to show to which extent the changes in RBP expression and alternative splicing in diabetic conditions are conserved between mice and humans.
2. In Fig. 1E and 1F it will be interesting to also highlight the other top pentamers enriched in T2D-regulated splicing events and whether they overlap with known RBP binding motifs.
3. The cut-off of 1% difference in PSI in the differential splicing analyses is a bit too low in my opinion and may capture some transcriptomic noise, although the overlap between Rbfox2-mut islets and KD MIN6 is remarkable. I suggest increasing it to at least 5%. Is the overlap between T2D and Rbfox2-mut maintained at higher Δ PSI cut-offs?
4. The authors mention that the oral glucose tolerance test showed no difference between wild-type and Rbfox-mut mice. This seems odd with the glucose intolerance detected with intraperitoneal injection. Can the authors please explain this apparent discrepancy?
5. The authors mention that Rbfox2 depletion alters insulin secretion by mainly affecting the first phase of insulin release, as indicated by IP-GTT. This is a very relevant point since later in the manuscript the authors link this with the observed reduction in docked granules in Rbfox2-mut beta cells, and with the general hypothesis that Rbfox2-regulated splicing changes in SNARE proteins may contribute to the disruption in the first phase observed in diabetes. To reinforce this hypothesis, the authors could provide additional evidences that Rbfox2 depletion alters glucose-stimulated insulin secretion mainly by affecting the first phase, for example by performing dynamic insulin secretion experiments or by static secretion experiments using KATP channel inhibitors.

6. In line with the previous point, to reinforce the proposed hypothesis, it would be interesting to test whether skipping of Rbfox2-regulated exons, for instance in Snap25, Syt7 or Stxbp1, using antisense oligonucleotides impairs insulin secretion in beta cells.

7. In the eCLIP analysis, it would be useful to show the k-mers enriched in Rbfox2 binding sites and how they compare to previous Rbfox CLIP experiments.

8. In the supplementary tables describing alternatively spliced events, it would be useful to provide the genomic coordinates of each event.

Jonàs Juan-Mateu
Centre for Genomic Regulation (CRG)
Barcelona, Spain

Reviewer #2 (Remarks to the Author):

The manuscript by Moss et al describes the study of splicing regulator RBFOX2 in insulin-producing islet cells, diving both into the phenotypic results of RBFOX2 manipulation as well as genomics views of RNA interactions and splicing changes controlled by RBFOX2. The biological insights (although not my area of expertise) I think are quite exciting, and the -omics data is well-described and provides significant insights into the mechanisms of how RBFOX2 is altering RNA processing

My only big-picture comment is that I think some of the splicing RNA-seq & CLIP analysis would be more informative at the event- rather than gene-level, as the mechanism of how an RBFOX2 eCLIP peak that isn't in the introns adjacent to the alternative exon is very unclear.

E.g. Fig 3E/Sup Fig 3D – I would find this analysis more informative at the event- rather than gene-level – particularly as the overlap between mut and KD is surprisingly strong. Something like an event-level scatter plot of the splicing change (delta-PSI value) between mouse islet and MIN6 would be quite interesting to see exactly how correlated the splicing changes are (and may help explain why there are surprisingly zero events seen only in the KD). Similarly, for Fig 5a, what does this look like only considering peaks in the introns flanking the alternative exon?

Relatedly, additional details on the CLIP analysis side would be helpful - e.g. what is the significance cutoff being used to define 'peaks', 'bound', etc. Figure 5A seems to suggest that 'bound' in that figure means gene-level enrichment, which I don't think is a very useful criteria to use for an RBP that tends to have specific small intronic peaks.

I'd also like to see some additional details on the splicing maps in Fig 5D-F; unless I'm mistaken I don't see details on how many 'no change' events make up the background set (or really how they were picked; I assume it's the same as the description in the methods for selecting background events for the motif analysis, but even there it's not really described in detail), but it would also be good to see some sort of statistic or variance analysis to say whether the differences seen are significant or noise (possibly a bootstrapping / random sampling kind of approach to see if the difference seen is observed with random equally sized sets of non-changed exons?)

Obviously having only 1 replicate of CLIP data is a limitation here; having two replicates may not be essential but would significantly improve the robustness of the analysis and interpretation.

Additionally, in the discussion – “Additionally, the Gsn gene does not contain the RBFOX consensus

sequence (GCAUG)" – was 'gene' supposed to mean 'the differential exon in Gsn'? The Gsn gene definitely contains multiple GCATG sequences (including the rat genome in introns flanking the Gsn alternative exon described in Jaun-Mateu et al), so either this is wrong or very unclearly written.

Other comments:

This isn't essential, but I think mechanistically it would be quite interesting to know whether for the events listed in Fig 6A-B RBFOX2 is only binding them in islet cells, or whether those interactions occur in other cell types as well; even just seeing whether some of those binding events seen in CLIP data from ENCODE or other studies (e.g. the brain RBFOX2 CLIP data from PMID: 24613350 or 24213538) might be quite informative (as well as validate the interactions seen here). An even bigger-picture view of how different RBFOX2 binding is between islets and brain would be very interesting, though likely outside the scope of this manuscript

Having bigwig files deposited at GEO as well as the peaks available there now would be useful in having readers be able to make rapid use of the data described here

Minor comments:

- 1) There's some typos in the GEO submission (the RBFOX2 is confusingly labeled 'RBFOX2_IgG', and the input I think the I at the beginning is a typo from 'gG' later?);
- 2) Sup Fig 5d, typo 'Sequencign'

Reviewer #3 (Remarks to the Author):

In this work Moss and co-workers investigate the role of RBFOX2 on insulin secretion. RBFOX2 is an RNA-binding protein (RBP). Dysregulation of RBPs through the following mRNA splicing occurs at the onset of diabetes. The focus of this manuscript is on how RBFOX2, through alternative splicing of genes required for insulin granules docking and exocytosis, regulates insulin secretion. The authors present some human data, but mainly make use of pancreas-specific Rbfox2-mutant mouse and Rbfox2-KD MIN6 cells to show that splicing of SNARE-proteins effects insulin secretion. This is novel and interesting data. The work is well-written and contain really good cartoons to make it easy to follow the authors thinking. The results contribute to increased understanding why exocytotic genes have reduced expression in T2D. The authors refer to previous studies in the field and the work presented is a natural next step understanding effects of RBPs on insulin secretion. However, the conclusions and claims are not fully supported by the data and therefore additional evidence are needed. There are some flaws in the data analysis mainly due to low n-values and the methodologies are good but some key experiments are missing. Hence, in the current form the interpretations are overexaggerated. Below you find details of my main concerns

1. Need to bring up the statistic as one of the major drawbacks of this study. The authors need to improve their n-values of some of the experiment to make them trustworthy. Some parts of the study have N=2 and N=1 which is definitely not good enough. Key experiments should have more than N=3. Also, data in the main figures are presented as a mix of male, female, and both together. A stricter line of presentation are needed. In some cases, there seems to be sex-differences and in some not at all. This is not totally clear.
2. Fig 3. Cacna1a is one of the genes effected and Ca²⁺ dependent exocytosis is top of processes suggested to be affected by RBFOX2 splining. However, there is no investigation on how the influx of Ca²⁺ is affected. The authors should investigate effects on Ca²⁺-currents and/or intracellular Ca²⁺ levels and investigate splicing effects also on Cacna1a.

3. Fig 4 present glucose-stimulated insulin secretion (N=3) on female mice only! This is a discrepancy of Suppl Fig 4 where the combined (male and female) glucose-stimulated insulin secretion is not significantly reduced, only K⁺ induced secretion. This needs to be better explained and foremost if the female data should be presented in the main part of the manuscript the number of experiments should be increased. This is also an example of what is mentioned in point #1 that data presented in the main figures are chosen, sometimes female (Fig 4A), sometimes male (e.g. Fig 2G, H) and sometimes a mix of both sexes (Fig 4B-E). The authors need to present either both male and female if they believe there are differences OR a mix of male /female in the main figures with the separate male and female data in the suppl. If there are sex-differences this should be discussed.

4. Fig 4. The exact number of docked granules per cell is not totally clear from the analyses. It is therefore hard to judge if it really is a reduction in the number docked granules. What is the number granules per cell-volume and number of docked granules per surface area?

5. Although authors show reduced K⁺-induced insulin secretion, exocytosis also need to be investigated with more specific methods such as TIRF-imaging or capacitance measurements.

MINOR:

1. Presentation of data should show individual data-points together with the bar-graphs.

2. What is the unit of Fig 4E. The figure needs to be better explained. Is it surface density? Why no variation?

Response to Reviewers

We appreciate the reviewers' critical evaluation and overall enthusiasm for this study. We have experimentally addressed their comments; the additional experiments complement our original findings and strengthen our overall conclusions. We summarize our response to the reviewers and changes to the manuscript below.

Reviewer #1 (Remarks to the Author):

The regulation and function of alternative splicing in pancreatic islets and its role in diabetes is an exciting topic that remains relatively understudied. In this manuscript, Moss, Sussel and colleagues provide multiple evidences that the RNA-binding protein RBFOX2 plays an important role in pancreatic beta cell function by regulating the alternative splicing of genes involved in insulin granule docking and exocytosis. Using a conditional islet-specific Rbfox2 mutant mice, the authors show that lack of Rbfox2 causes reduced glucose-stimulated insulin secretion and glucose intolerance. Combining RNAseq analysis of Rbfox2 mutant mouse islets and eCLIP experiments in the MIN6 beta cell line, the authors identified hundreds of alternative splicing events directly regulated by Rbfox2 in genes relevant for insulin release, particularly in genes involved in granule docking and exocytosis. In line with this, using electron microscopy the authors show that Rbfox2 depletion causes alterations in insulin granule organization and a reduction in the number of docked granules to the plasma membrane. Interestingly, the authors also provide evidences that Rbfox2 is mis-regulated in diabetic conditions, correlating with splicing mis-regulation in T2D islets, and propose the exciting hypothesis that these splicing changes may contribute to the defects in the first phase of insulin secretion observed in T2D patients.

Overall, this is a novel and very interesting work that demonstrates a cellular and physiological role of Rbfox2-regulated splicing in pancreatic beta cell function and glucose homeostasis, expanding the current knowledge about the roles of alternative splicing in islets.

I have only minor comments:

1. The authors show in Figure 1 that RBFOX2 is down-regulated in diabetic conditions and that this correlates with alternative splicing changes observed in T2D. While the data is convincing in mice, is somehow weaker in humans probably due to the low number of samples analysed. Data from a larger number of donors (both for bulk and single cell RNAseq) is publicly available and would enable them to increase the power and robustness of their analysis. Additionally, it would be interesting to show to which extent the changes in RBP expression and alternative splicing in diabetic conditions are conserved between mice and humans.

We totally agree with the reviewer. Unfortunately the read depth for the majority of available human datasets is not sufficient to identify alternatively spliced transcripts. In addition most of the data sets used short read sequencing that does not accurately capture isoform usage. Due to the relevance of this data, but lack of statistical power, we have moved this data to supplemental Figure S2C.

We also like the idea of comparing mouse and human splicing events, however, there is usually a difference between exon positions, the datasets are not directly comparable.

2. In Fig. 1E and 1F it will be interesting to also highlight the other top pentamers enriched in T2D-regulated splicing events and whether they overlap with known RBP binding motifs.

Figure 1C and Figure S2D identify kmer enrichment downstream of exons that are included in control samples and skipped in samples from diabetic donors. In both mouse and human datasets, we observe significant enrichment of the RBFOX2 binding sequence GCAUG. We have included new supplemental tables that list other top significantly enriched 5-mers around. The addition of this comparison identified the RBFOX2 consensus sequence as one of only seven k-mers enriched at the same position between the mouse and human datasets.

3. The cut-off of 1% difference in PSI in the differential splicing analyses is a bit too low in my opinion and may capture some transcriptomic noise, although the overlap between Rbfox2-mut islets and KD MIN6 is remarkable. I suggest increasing it to at least 5%. Is the overlap between T2D and Rbfox2-mut maintained at higher Δ PSI cut-offs?

The 1% PSI cut-off was selected based on PSI cut-off values published in other fields. However, as recommended, we redid the analyses using an increased Δ PSI of 5% and report the findings in Figure S5. We observe and maintain overlap of 112 splicing sensitive genes across datasets. At 5% Δ PSI we also maintain many of the enriched GO Terms. In addition, we compared increasing Δ PSI percentages in the overlap of the Rbfox2-mut and Rbfox2-KD datasets and consistently observe a high degree of overlap of Rbfox2 splicing sensitive genes. This is likely due to the fact that the majority of cells in the mouse islet are β cells. Furthermore this shows the robustness of our two Rbfox2 models in vivo and in vitro. There is also still substantial overlap between T2D and Rbfox2-mut using the higher Δ PSI cut-off.

4. The authors mention that the oral glucose tolerance test showed no difference between wild-type and Rbfox-mut mice. This seems odd with the glucose intolerance detected with intraperitoneal injection. Can the authors please explain this apparent discrepancy?

While both tests are measures of glucose tolerance, the IP-GTT assesses first phase insulin secretion through direct glucose stimulation while the O-GTT assesses second phase insulin secretion through GLP-1 signaling from the intestine. The two tests are commonly used to distinguish which pathways are defective and do not always correlate if only one pathway is affected. An additional line of text in the results section has clarified the differences in the IP-GTT and O-GTT.

5. The authors mention that Rbfox2 depletion alters insulin secretion by mainly affecting the first phase of insulin release, as indicated by IP-GTT. This is a very relevant point since later in the manuscript the authors link this with the observed reduction in docked granules in Rbfox2-mut beta cells, and with the general hypothesis that Rbfox2-regulated splicing changes in SNARE proteins may contribute to the disruption in the first phase observed in diabetes. To reinforce this hypothesis, the authors could provide additional evidences that Rbfox2 depletion alters glucose-stimulated insulin secretion mainly by affecting the first phase, for example by performing dynamic insulin secretion experiments or by static secretion experiments using KATP channel inhibitors.

We have performed the dynamic insulin secretion experiments (dGSIS) as suggested. The results are presented in Figure 4D. The results of the assay confirm the impairment of insulin secretion in the Rbfox2 mutant islets. Since we were not equipped to perform these experiments in our lab, we enlisted the expertise of Dr. Patrick Macdonald at the U. of Alberta. Dr. Macdonald and his trainees are now included as authors to acknowledge their contribution.

6. In line with the previous point, to reinforce the proposed hypothesis, it would be interesting to test

whether skipping of Rbfox2-regulated exons, for instance in Snap25, Syt7 or Stxbp1, using antisense oligonucleotides impairs insulin secretion in beta cells.

Previously published studies have investigated Syt7, Stxbp1, and Snap25 gene and isoform expression (PMID: 26216970, PMID: 28798351, PMID: 10580425). Disruption of each gene has the potential to dysregulate glucose stimulated insulin secretion independently. Due to this fact, we would hypothesize that mimicking the individual splicing defects would produce results similar to the previously conducted genetic experiments. Furthermore, we would not predict that rescue of an individual SNARE complex gene by ASO would rescue the complete Rbfox2-mut phenotype due to the combinatorial impact of multiple splicing defects across several glucose stimulated insulin secretion pathway genes.

7. In the eCLIP analysis, it would be useful to show the k-mers enriched in Rbfox2 binding sites and how they compare to previous Rbfox CLIP experiments.

6-mer enrichment under RBFOX2-eCLIP peaks was determined and the previously characterized UGCAUG sequence from other published experiments is among the most highly enriched 6-mers under these peaks. This information has been added to the text. The top 7 enriched 6-mers are listed in Figure Figure S7B, S8A-B.

8. In the supplementary tables describing alternatively spliced events, it would be useful to provide the genomic coordinates of each event.

The genomic coordinates for each splicing event type have been included as additional supplemental tables.

Reviewer #2 (Remarks to the Author):

The manuscript by Moss et al describes the study of splicing regulator RBFOX2 in insulin-producing islet cells, diving both into the phenotypic results of RBFOX2 manipulation as well as genomics views of RNA interactions and splicing changes controlled by RBFOX2. The biological insights (although not my area of expertise) I think are quite exciting, and the -omics data is well-described and provides significant insights into the mechanisms of how RBFOX2 is altering RNA processing

My only big-picture comment is that I think some of the splicing RNA-seq & CLIP analysis would be more informative at the event- rather than gene-level, as the mechanism of how an RBFOX2 eCLIP peak that isn't in the introns adjacent to the alternative exon is very unclear.

We apologize for the confusion. We have clarified in the text when an analysis was done on the exon level vs. when it is done on the gene level.

E.g. Fig 3E/Sup Fig 3D – I would find this analysis more informative at the event- rather than gene-level – particularly as the overlap between mut and KD is surprisingly strong. Something like an event-level scatter plot of the splicing change (delta-PSI value) between mouse islet and MIN6 would be quite interesting to see exactly how correlated the splicing changes are (and may help explain why there are surprisingly zero events seen only in the KD). Similarly, for Fig 5a, what does this look like only considering peaks in the introns flanking the alternative exon?

Relatedly, additional details on the CLIP analysis side would be helpful - e.g. what is the significance

cutoff being used to define 'peaks', 'bound', etc. Figure 5A seems to suggest that 'bound' in that figure means gene-level enrichment, which I don't think is a very useful criteria to use for an RBP that tends to have specific small intronic peaks.

Additional details about eCLIP-Seq analysis and peak calling have been added to the *eCLIP Sequencing and Analysis* section of the methods and language in the figure legend for 5A has been updated. In Figure 5A the "RBFox2-bound genes" are genes with multiple significant RBFox2 eCLIP peaks that were identified by clipper peak calling analysis. This list of genes was used in Figure 5A to narrow down the list of RBFox2 splicing sensitive genes to those that are potentially directly regulated by RBFox2 binding. In Figure 5, only the first Venn diagram is at the gene level, all other data in the figure is presented at the exon level.

I'd also like to see some additional details on the splicing maps in Fig 5D-F; unless I'm mistaken I don't see details on how many 'no change' events make up the background set (or really how they were picked; I assume it's the same as the description in the methods for selecting background events for the motif analysis, but even there it's not really described in detail), but it would also be good to see some sort of statistic or variance analysis to say whether the differences seen are significant or noise (possibly a bootstrapping / random sampling kind of approach to see if the difference seen is observed with random equally sized sets of non-changed exons?)

The original figure included a dashed black line showing enrichment of RBFox2 binding events near "insensitive exons", or defined cassette exons (in the case of the now Figure 5B-E) that are not significantly alternatively spliced in the Rbfox2-KD dataset. In response to this comment, we have used bootstrapping of RBFox2 peaks at these insensitive splicing events to identify the probability of RBFox2 binding and determined statistical significance using a Poisson distribution, p-values are indicated on the plot.

Obviously having only 1 replicate of CLIP data is a limitation here; having two replicates may not be essential but would significantly improve the robustness of the analysis and interpretation.

A second replicate of the RBFox2-eCLIP experiment was conducted and data presented in the main figure reflects both replicates. The replicates are individually compared in Figure S8. The replicates produced similar results and the conclusions are unchanged.

Additionally, in the discussion – “Additionally, the Gsn gene does not contain the RBFox consensus sequence (GCAUG)” – was 'gene' supposed to mean 'the differential exon in Gsn'? The Gsn gene definitely contains multiple GCATG sequences (including the rat genome in introns flanking the Gsn alternative exon described in Jaun-Mateu et al), so either this is wrong or very unclearly written.

The text for this section has been revised for clarity. There are no significant RBFox2-eCLIP peaks in the Gsn gene suggesting that despite the presence of the UGCAUG sequence in the gene none of them are occupied by RBFox2. Furthermore, based on our Rbfox2-KD and Rbfox2-mut data sets, it appears that Gsn transcript splicing is also not dependent on RBFox2 expression.

Other comments:

This isn't essential, but I think mechanistically it would be quite interesting to know whether for the events listed in Fig 6A-B RBFox2 is only binding them in islet cells, or whether those interactions occur in

other cell types as well; even just seeing whether some of those binding events seen in CLIP data from ENCODE or other studies (e.g. the brain RBFOX2 CLIP data from PMID: 24613350 or 24213538) might be quite informative (as well as validate the interactions seen here). An even bigger-picture view of how different RBFOX2 binding is between islets and brain would be very interesting, though likely outside the scope of this manuscript

While a thorough cross cell type comparison of RBFOX2 binding and RNA regulation is outside the scope of this study, we would agree that it is an exceptionally interesting question. We explored the recommended datasets and found overlapping RBFOX2 binding sites in 1,853 genes, including *Stxbp1*, *Syt7* and *Snap25* from the HITS-CLIP experiment in mouse neurons (PMID: 24613350). Moreover, we find that there is overlap with 2341 of our 6514 RBFOX2 significant eCLIP genes. These findings are not surprising considering the functional redundancy of SNARE complex components in the secretion of insulin from the β cell and neurotransmitter from neurons. However, there are several differences in these studies (including the type of CLIP experiment, peak calling, and filtering) that may not make them directly comparable. Cardiac cells express a different set of RBFOX2 isoforms, and therefore may have different targets. These are studies we are continuing to pursue.

Having bigwig files deposited at GEO as well as the peaks available there now would be useful in having readers be able to make rapid use of the data described here

BigWig files have been deposited on GEO and are associated with Series GSE221276.

Minor comments:

1) There's some typos in the GEO submission (the RBFOX2 is confusingly labeled 'RBFOX2_IgG', and the input I think the I at the beginning is a typo from 'gG' later?);

The referenced errors have been corrected in GEO.

2) Sup Fig 5d, typo 'Sequencign'

Corrected

Reviewer #3 (Remarks to the Author):

In this work Moss and co-workers investigate the role of RBFOX2 on insulin secretion. RBFOX2 is an RNA-binding protein (RBP). Dysregulation of RBPs through the following mRNA splicing occurs at the onset of diabetes. The focus of this manuscript is on how RBFOX2, through alternative splicing of genes required for insulin granules docking and exocytosis, regulates insulin secretion. The authors present some human data, but mainly make use of pancreas-specific *Rbfox2*-mutant mouse and *Rbfox2*-KD MIN6 cells to show that splicing of SNARE-proteins effects insulin secretion.

This is novel and interesting data. The work is well-written and contain really good cartoons to make it easy to follow the authors thinking. The results contribute to increased understanding why exocytotic genes have reduced expression in T2D. The authors refer to previous studies in the field and the work presented is a natural next step understanding effects of RBPs on insulin secretion.

However, the conclusions and claims are not fully supported by the data and therefore additional evidence are needed. There are some flaws in the data analysis mainly due to low n-values and the

methodologies are good but some key experiments are missing. Hence, in the current form the interpretations are overexaggerated. Below you find details of my main concerns

1. Need to bring up the statistic as one of the major drawbacks of this study. The authors need to improve their n-values of some of the experiment to make them trustworthy. Some parts of the study have N=2 and N=1 which is definitely not good enough. Key experiments should have more than N=3. Also, data in the main figures are presented as a mix of male, female, and both together. A stricter line of presentation are needed. In some cases, there seems to be sex-differences and in some not at all. This is not totally clear.

All experiments (including those in main and supplemental figures) that were performed for this study and included in this manuscript contained an n = 3 or more samples. The n = 1 and n = 2 that the reviewer has referred to comes from a separately published study from an independent group using rare sorted human β cells from islet donors with and without T2D. Because the available datasets of human islet samples are variable in their availability, quality, and sequencing depth, we were not able to find additional datasets to mine for this study. Due to the relevance of this data but lack of statistical power, we have moved this data to Supplemental Figure S2C.

2. Fig 3. *Cacna1a* is one of the genes effected and Ca^{2+} dependent exocytosis is top of processes suggested to be affected by *RbFOX2* splicing. However, there is no investigation on how the influx of Ca^{2+} is affected. The authors should investigate effects on Ca^{2+} -currents and/or intracellular Ca^{2+} levels and investigate splicing effects also on *Cacna1a*.

Capacitance measurements were performed for both control and *Rbfox2*-mut β cells. The results presented in Figure 4I indicate that despite significantly decreased insulin granule exocytosis, there were no differences in calcium current (Figure S6H). Since we were not equipped to perform these experiments in our lab, we enlisted the expertise of Dr. Patrick Macdonald at the U. of Alberta. Dr. Macdonald and his trainees are now included as authors to acknowledge their contribution.

3. Fig 4 present glucose-stimulated insulin secretion (N=3) on female mice only! This is a discrepancy of Suppl Fig 4 where the combined (male and female) glucose-stimulated insulin secretion is not significantly reduced, only K^+ induced secretion. This needs to be better explained and foremost if the female data should be presented in the main part of the manuscript the number of experiments should be increased. This is also an example of what is mentioner in point #1 that data presented in the main figures are chosen, sometimes female (Fig 4A), sometimes male (e.g. Fig 2G, H) and sometimes a mix of both sexes (Fig 4B-E). The authors need to present either both male and female if they believe there are differences OR a mix of male /female in the main figures with the separate male and female data in the suppl. If there are sex-differences this should be discussed.

The phenotypic data displayed in Figure 2 is from male *Rbfox2*-mut mice and their corresponding controls. Additional timepoints and data collected from female mice are now presented in Figure S3. Similarly, Figure 5 presents dGSIS data collected in male mice but the results are confirmed in complementary static GSIS in both male and female mice in Figure S6E-F. We observe no overt differences between the male and female phenotypes. This information has been added to the text.

4. Fig 4. The exact number of docked granules per cell is not totally clear from the analyses. It is therefore hard to judge if it really is a reduction in the number docked granules. What is the number granules per cell-volume and number of docked granules per surface area?

In Figure 2C-E we show that there are no differences in Ins1 or Ins2 transcript expression, INS or c-peptide 1 protein expression (by IF). This data indicates that there is sufficient insulin being produced by the cells and therefore we did not quantify total insulin granule density. We quantified insulin granule size and distance from the plasma membrane in defined subsections of the β cell. In Supplemental Figure 6A we are careful to show that we quantified the same average cell area across images and that the average distance from the edge of the image to the plasma membrane was not different (Figure S6B). Additionally, we note in the Methods that we quantified and measured approximately the same number of granules for each genotype across the biological replicates (700 and 657 granules respectively). Additionally, we have added Supplemental Figure 6D that shows there is no difference in insulin granule density across the quantified region of each β cell.

5. Although authors show reduced K⁺-induced insulin secretion, exocytosis also need to be investigated with more specific methods such as TIRF-imaging or capacitance measurements.

Capitance measurements were performed for both control and Rbfox2-mut β cells. The results presented in Figure 4I indicate that despite significantly decreased insulin granule exocytosis, there were no differences in calcium current (Figure S6H). Since we were not equipped to perform these experiments in our lab, we enlisted the expertise of Dr. Patrick Macdonald at the U. of Alberta. Dr. Macdonald and his trainees are now included as authors to acknowledge their contribution.

MINOR:

1. Presentation of data should show individual data-points together with the bar-graphs.

Individual datapoints have been added to the bar-graphs where relevant and are referenced in the figure legends.

2. What is the unit of Fig 4E. The figure needs to be better explained. Is it surface density? Why no variation?

Figure 4E was corrected to include the p-value ($p = 0.007$) comparing the proportions across samples from the Fisher's Exact test comparing counts of docked vs. undocked insulin granules across three biological replicate mice described in the figure legend.

REVIEWERS' COMMENTS

Reviewer #1 (Remarks to the Author):

I appreciate the time and effort that the authors took to address all my concerns and suggestions with new experiments and analyses. I have no additional comments.

Reviewer #2 (Remarks to the Author):

I appreciate the authors' addressing of my comments and I think this is an exciting result!

As one final comment, in the revision I'm still confused by some of the overlap analysis – ie Fig. S5A says there are 0 MIN6-specific, but then Fig. S5B (the 2nd one, should be C) seems to have yellow points that indicate significance in MIN6 but not Islet? (Is S5B supposed to be a different KD B cells instead of MIN6?). I can't quite tell if this just needs better language to define what's different between these figures or if it's because the cutoffs are too low, but I'm still surprised by seeing '0' dataset-specific events and I'm having difficulty understanding how those figures correspond with each other

Other notes:

Fig. 2 sub-panel labels are wrong (H is before G, I is missing)

Pg 7, line 174 typo (Figure 4SA should be S4A)

Fig S5 has two B subpanels

I know I've seen it both ways in the literature, but to me since both Cassette Exon and Constitutive Exon can be abbreviated 'CE', I think it's a bit more helpful to the reader to do it as SE (skipped exon) or AE (alternative exon)

I'm a bit confused by Syt2's inclusion in Fig. 6A-B, since it's only in 1 dataset and doesn't seem to have RBFOX2 CLIP peaks

I think the Methods section ('STAR Methods') is still formatted for Cell family journals

Reviewer #3 (Remarks to the Author):

My misunderstanding concerning n-values and female/male questions are now clearly described and the authors have made a good work to explain and better present this information and the connected data. I want to emphasize that in all this work present novel and interesting data and the authors have made reasonable updates of the manuscript based on the raised comments. I only have some minor concerns to the current new version.

1. I am happy that the authors now have performed capacitance measurements to investigate exocytosis. Indeed, the effect might be stronger than seen do to that the insulin granules become bigger in the KO-mice. For a better understanding I suggest that the authors present example traces from the capacitance measurements. Moreover, it is not clear from the results/figure legend/methods if it was a single or multiple depolarization(s). Between which voltages was the depolarization performed? How long was the depolarization?

2. It is strange that there is no change in the Ca-current in the KO although CACNA1A is one of the genes regulated by RBFOX2. CACNA1A is the main Ca²⁺ channel for exocytosis in mouse β -cells. This

observation at least deserves some discussion around why? Can it perhaps be explained by that other voltage-sensitive Ca²⁺ channel genes are upregulated as a compensation? The best would be to perform Ca²⁺ measurements and/or more specific Ca²⁺-current measurements (without glutamate) to investigate this further with more specific methods, but I do understand that this might be out of the scope of this manuscript.

3. In my version the line square around the grey middle bar in figure S4B (RBFOX2 western blot quantification) is smaller than the field grey area.

Response to Reviewers

Reviewer #1 (Remarks to the Author):

I appreciate the time and effort that the authors took to address all my concerns and suggestions with new experiments and analyses. I have no additional comments.

We appreciate the reviewer's support

Reviewer #2 (Remarks to the Author):

I appreciate the authors' addressing of my comments and I think this is an exciting result!

As one final comment, in the revision I'm still confused by some of the overlap analysis – ie Fig. S5A says there are 0 MIN6-specific, but then Fig. S5B (the 2nd one, should be C) seems to have yellow points that indicate significance in MIN6 but not Islet? (Is S5B supposed to be a different KD B cells instead of MIN6?). I can't quite tell if this just needs better language to define what's different between these figures or if it's because the cutoffs are too low, but I'm still surprised by seeing '0' dataset-specific events and I'm having difficulty understanding how those figures correspond with each other

Apologies for the confusion. The data in figure S5A represents data at the **gene** level while the S5C shows data at the **exon** level. For this reason, many of the red and yellow points (which can represent several exons in a single gene) in figure S5C collapse into a single gene in figure S5A. We have added additional text to help clarify this point.

Other notes:

Fig. 2 sub-panel labels are wrong (H is before G, I is missing).

Corrected.

Pg 7, line 174 typo (Figure 4SA should be S4A)

Corrected.

Fig S5 has two B subpanels

Corrected.

I know I've seen it both ways in the literature, but to me since both Cassette Exon and Constitutive Exon can be abbreviated 'CE', I think it's a bit more helpful to the reader to do it as SE (skipped exon) or AE (alternative exon)

The original label of CE (cassette exon) was selected to denote both inclusion and skipped exon of the cassette exon type. For clarity in the text we have changed the terms from cassette exon (CE) to cassette/skipped exon (SE).

I'm a bit confused by Syt2's inclusion in Fig. 6A-B, since it's only in 1 dataset and doesn't seem to have RBFOX2 CLIP peaks

Syt2 was originally included because it was identified in the MIN6 alternative splicing analysis described in Figure 3. It has been removed from Figure 6 for clarity.

I think the Methods section ('STAR Methods') is still formatted for Cell family journals

We've changed the title of the Methods sections and reformatted the Key Resources table into Supplemental Table 19 to conform to Nature Communication formatting.

Reviewer #3 (Remarks to the Author):

My misunderstanding concerning n-values and female/male questions are now clearly described and the authors have made a good work to explain and better present this information and the connected data. I want to emphasize that in all this work present novel and interesting data and the authors have made reasonable updates of the manuscript based on the raised comments. I only have some minor concerns to the current new version.

1. I am happy that the authors now have performed capacitance measurements to investigate exocytosis. Indeed, the effect might be stronger than seen do to that the insulin granules become bigger in the KO-mice. For a better understanding I suggest that the authors present example traces from the capacitance measurements. Moreover, it is not clear from the results/figure legend/methods if it was a single or multiple depolarization(s). Between which voltages was the depolarization performed? How long was the depolarization?

We have added sample traces to supplemental figure 6I. Methods have been updated to include additional assay details "To measure exocytotic response as increases in cell surface area (capacitance) cells were held at -70 mV and subjected to a series of ten 500 ms depolarizations to 0 mV. Total exocytotic responses were taken at the difference between cell capacitance before depolarization and after the 10th pulse (in fF), and normalized to initial cell size (in pF)."

2. It is strange that there is no change in the Ca-current in the KO although CACNA1A is one of the genes regulated by RBFOX2. CACNA1A is the main Ca²⁺ channel for exocytosis in mouse β-cells. This observation at least deserves some discussion around why? Can it perhaps be explained by that other voltage-sensitive Ca²⁺ channel genes are upregulated as a compensation? The best would be to perform Ca²⁺ measurements and/or more specific Ca²⁺-current measurements (without glutamate) to investigate this further with more specific methods, but I do understand that this might be out of the scoop of this manuscript.

Because the CACNA1A gene contains 47 exons, many of which are subject to alternative splicing, the number of CACNA1A splice isoforms is estimated to be in the order of thousands, many of which are functional. Therefore, it is possible that the altered protein isoforms of CACNA1 resulting from the loss of *Rbfox2* do not affect protein function or that the relative ratio of isoforms is not sufficiently altered to cause a phenotype. Future studies will explore the potential implications of alternative splice variants in the *Cacna1A* gene. This information has been added to the discussion.

3. In my version the line square around the grey middle bar in figure S4B (RBFOX2 western blot quantification) is smaller than the field grey area.

This issue seems to occur when the Ai file is converted to a PDF. This has been corrected.